# An hepatitis B and D virus infection model using human pluripotent stem cell-derived hepatocytes

Huanting Chi [1,2,10], Bingqian Qu [1,3,7,10], Angga Prawira[3], Talisa Richardt[3], Lars Maurer[1,4], Jungen Hu[1], Rebecca M Fu[1], Florian A Lempp [3,8], Zhenfeng Zhang[3,9], Dirk Grimm [2,4,5], Xianfang Wu [6], Stephan Urban [2,3,11✉] & Viet Loan Dao Thi [1,2,11✉]

## Abstract

**Current culture systems available for studying hepatitis D virus (HDV) are suboptimal. In this study, we demonstrate that hepatocyte-like cells (HLCs) derived from human pluripotent stem cells (hPSCs) are fully permissive to HDV infection across various tested genotypes. When co-infected with the helper hepatitis B virus (HBV) or transduced to express the HBV envelope protein HBsAg, HLCs effectively release infectious progeny virions. We also show that HBsAg-expressing HLCs support the extracellular spread of HDV, thus providing a valuable platform for testing available anti-HDV regimens. By challenging the cells along the differentiation with HDV infection, we have identified CD63 as a potential HDV co-entry factor that was rate-limiting for HDV infection in immature hepatocytes. Given their renewable source and the potential to derive hPSCs from individual patients, we propose HLCs as a promising model for investigating HDV biology. Our findings offer new insights into HDV infection and expand the repertoire of research tools available for the development of therapeutic interventions.**

**Keywords** Hepatitis D Virus; Hepatitis B Virus; Human Pluripotent Stem Cells; Hepatocyte-like Cells; Antiviral Treatment
**Subject Categories** Methods & Resources; Microbiology, Virology & Host Pathogen Interaction; Stem Cells & Regenerative Medicine

## Introduction

Approximately 5% of chronic hepatitis B virus (HBV) carriers are co- or super-infected with hepatitis D virus (HDV), resulting in an estimated 12 million chronic hepatitis D (CHD) patients worldwide (Stockdale et al, 2020). However, a recent study suggests that these figures are likely to underestimate the true global burden of disease (Chen et al, 2019). HBV/HDV co-infection can cause the most aggressive form of viral hepatitis, leading to an accelerated progression of liver dysfunction and disease (Farci and Niro, 2012).

HDV is a circular, single-stranded, negative-sense RNA virus that belongs to the *Kolmioviridae* family. Based on their genome diversity, eight different HDV genotypes (GTs) have been described, each with a distinct geographical distribution (Lempp and Urban, 2017). HDV-1 is the most prevalent genotype and is endemic worldwide, while infections with genotypes HDV-2 to -8 occur regionally. HDV-2 infections are common in East and Southeast Asia, while the most divergent genotype, HDV-3, is found exclusively in South America. HDV-4 is found in Japan and Taiwan and HDV-5 to -8 infections are restricted to Africa (Le Gal et al, 2017). Infection with the different GTs can lead to different clinical outcomes: Compared to HDV-2, both HDV-1 and HDV-3 infections can lead to severe hepatitis with more adverse patient outcomes (Casey et al, 1993). Patients infected with HDV-4 usually experience a mild course of liver disease. Patients with an HDV-5 infection appear to have a preferable prognosis and respond better to pegylated interferon alpha (peg-IFN-α) treatment than HDV-1 patients (Spaan et al, 2020). The clinical features of recently identified genotypes HDV-6 to -8 are less characterized (Hughes et al, 2011).

HDV recruits HBV surface envelope proteins (HBsAg) for its progeny envelopment, which is critical for HDV propagation and transmission (Alfaiate et al, 2015). Similar to HBV, HBsAg-enveloped HDV virions enter hepatocytes through the interaction with the integral transmembrane protein receptor Na⁺-taurocholate co-transporting polypeptide (NTCP) (Yan et al, 2012). NTCP not only determines HDV liver tropism, but also its species specificity, since only human NTCP, but not mouse, rat, pig or macaque NCTP homologs, supports HDV entry (Lempp and Urban, 2017; Lempp et al, 2017; Li et al, 2014; Yan et al, 2013).

[1]Schaller Research Group, Department of Infectious Diseases, Virology, Heidelberg University, Medical Faculty Heidelberg, Heidelberg, Germany. [2]German Centre for Infection Research (DZIF), Partner Site Heidelberg, Heidelberg, Germany. [3]Molecular Virology, Department of Infectious Diseases, Heidelberg University, Medical Faculty Heidelberg, Heidelberg, Germany. [4]Department of Infectious Diseases, Virology, Section Viral Vector Technologies, University Hospital Heidelberg, Cluster of Excellence CellNetworks, BioQuant, Center for Integrative Infectious Diseases Research (CIID), Heidelberg, Germany. [5]German Center for Cardiovascular Research (DZHK), Partner Site Heidelberg, Heidelberg, Germany. [6]Infection Biology Program and Department of Cancer Biology, Lerner Research Institute, Cleveland Clinic Foundation, Cleveland, OH, USA. [7]Present address: Division of Veterinary Medicine, Paul-Ehrlich-Institut, Langen, Germany. [8]Present address: Humabs Biomed SA, A Subsidiary of Vir Biotechnology, Bellinzona, Switzerland. [9]Present address: School of Public Health and Emergency Management, School of Medicine, Southern University of Science and Technology, Shenzhen, China. [10]These authors contributed equally: Huanting Chi, Bingqian Qu. [11]These authors contributed equally: Stephan Urban, Viet Loan Dao Thi.
✉E-mail: Stephan.Urban@med.uni-heidelberg.de; VietLoan.DaoThi@med.uni-heidelberg.de

The HDV genome replicates in the nucleus by a double-rolling circle mechanism, generating intermediate antigenomic and genomic viral RNA strands (Sureau and Negro, 2016). The mRNA transcribed from the genomic RNA encodes only a single open reading frame (ORF) leading to the translation of two proteins, the small (S-HDAg) and the large (L-HDAg) hepatitis delta antigen (Kuo et al, 1989). Editing of the amber stop codon of the S-HDAg ORF in the antigenome by adenosine deaminase acting on RNA 1 (ADAR1) elongates the reading frame by 19–20 codons, driving the transition from S- to L-HDAg synthesis (Polson et al, 1996). Within the extended L-HDAg terminus, a cellular farnesyltransferase recognition site is modified, allowing the switch to virion assembly and release: while S-HDAg facilitates HDV genome replication, farnesylated L-HDAg inhibits it. L-HDAg, together with the S-HDAg and the HDV genome, initiates the formation of the ribonucleoprotein (RNP) complex. The farnesyl group facilitates RNP attachment to the endoplasmic reticulum membrane, where HBsAg is located and new HDV progeny can be assembled (Verrier et al, 2016).

To date, fundamental aspects of HDV biology remain poorly understood, including the molecular details of its life cycle, the apparent genotype-dependent disease severity, and the underlying mechanisms leading to accelerated liver dysfunction (Lempp and Urban, 2017). As a result, off-label peg-IFN-α has long been the only available treatment for HDV infection. However, it cannot be used in all patients, has only limited efficiency, and is often associated with significant side effects (Lempp and Urban, 2017; Wranke et al, 2015). Bulevirtide (BLV, brand name Hepcludex, formerly known as Myrcludex B), was recently granted conditional marketing authorization by the European Medicines Agency (Jachs et al, 2022). It is based on a synthetic pre-S1-derived myristoylated peptide that specifically binds to NTCP, thereby blocking HBV and HDV entry. Recent real-world studies have shown that BLV, as a monotherapy or in combination with peg-IFN-α, has profound effects on serum and liver HDV RNA, with possible curative potential in a subset of patients (Lampertico et al, 2022). The anti-HDV efficacy of the farnesyltransferase inhibitor Lonafarnib (LNF), which inhibits HDV progeny envelopment and therefore secretion, has been demonstrated in a phase 3 clinical trial when given in combination with either ritonavir or peg-IFN-α (Mascolini, 2023).

The gap in our understanding of the molecular biology of HDV and the lack of a curative treatment is partly due to the limitations of reproducible HDV cell culture systems that mimic the physiological state of hepatocytes in vivo. Primary human hepatocytes (PHH) are an attractive model for in vitro HDV studies. However, their limited availability, high donor-to-donor variability, difficulty in culturing and maintaining them for long periods, and limited ability to manipulate them genetically place severe limitations on reproducible and high-throughput HDV studies (Verrier et al, 2016). Upon plating, the susceptibility of PHH to HDV infection decreases rapidly due to the loss of hepatocyte polarization and NTCP expression (Heuschkel et al, 2021). On the other hand, widely available and highly reproducible hepatoma cells can only be rendered permissive to HDV infection upon induced or ectopic NTCP expression (Yan et al, 2012). We have recently engineered a HepG2-derived cell line that co-expresses HBsAg in addition to NTCP, allowing us to study HDV spread (Lempp et al, 2019). However, hepatoma cells retain

their transformed nature and members of the drug-metabolizing CYP450 family are only expressed at low levels in these cells (Rodríguez-Antona et al, 2002). The human liver bipotent progenitor cell line HepaRG becomes permissive to HDV infection upon differentiation into hepatocytes (Lucifora et al, 2020). While differentiated HepaRGs cells retain many characteristics of PHHs, they do not replicate all of them (Jetten et al, 2013).

Recently, numerous protocols have been developed to differentiate human-induced or embryonic pluripotent stem cells (hiPSC/ESC) into hepatocyte-like cells (HLCs) (reviewed in (Bove et al, 2021)). HLCs resemble PHHs and recapitulate physiological features, such as drug metabolism or innate immune response, better than hepatoma cells (Szkolnicka et al, 2014). In addition, stem cells are a reproducible and renewable resource that can be genetically manipulated. We and others have shown that hESC- or iPSC-derived HLCs are suitable for HBV (Xia et al, 2019), hepatitis C virus (HCV) (Wu et al, 2012) and hepatitis E virus (HEV) (Wu et al, 2018) studies.

Here, we show that HLCs are readily permissive for HDV infection and replication of all genotypes tested. We found that the co-infection with HBV or adeno-associated virus (AAV)-mediated transduction of HBsAg in HLCs enables the assembly and release of HDV progeny. We demonstrate that HDV can spread extracellularly in HLCs, which led us to evaluate the efficacy of available anti-HDV regimens in HLCs. Finally, by challenging the cells at different stages of HLC differentiation, we highlight the potential of this system to identify novel host factors that either promote or restrict HDV infection.

## Results

### Hepatocyte-like cells (HLCs) are permissive for HDV infection in an NTCP-dependent manner

Previous studies have demonstrated the utility of stem cell-derived HLCs for hepatotropic virus infection, including HBV (reviewed in (Bove et al, 2021)). Here, we analyzed whether HLCs are also permissive to HDV infection. We first used the HDV genotype (GT) 1 strain (JC126) (Gudima et al, 2002) to infect HLCs that we had differentiated from the human embryonic stem cell (hESC) line WA09 using a previously optimized protocol (Dao Thi et al, 2020) (Fig. 1A). At the end of the differentiation, HLCs supported hepatocyte characteristics including indocyanine green uptake, glycogen synthesis, and the conversion of non-fluorogenic carboxyfluorescein diacetate into fluorogenic carboxyfluorescein (Dao Thi et al, 2020).

As shown in Fig. EV1A,B, we detected increasing numbers of HDAg-positive HLCs in an inoculum dose-dependent manner. HDV infection could be completely blocked by pre-treating HLCs with the inhibitor bulevirtide (BLV) (Fig. 1A), which is based on a synthetic lipopeptide that mimics the receptor binding site of the HBV pre-S1 protein and binds to the HBV/HDV entry receptor NTCP. Notably, we observed the formation of a second, highly confluent HLC population during differentiation of the cells in the final maturation medium. Although we detected some HDAg-positive cells in the less confluent HLC population, we found the majority of HDV infections in the highly confluent HLC population (Figs. 1A and EV1A).

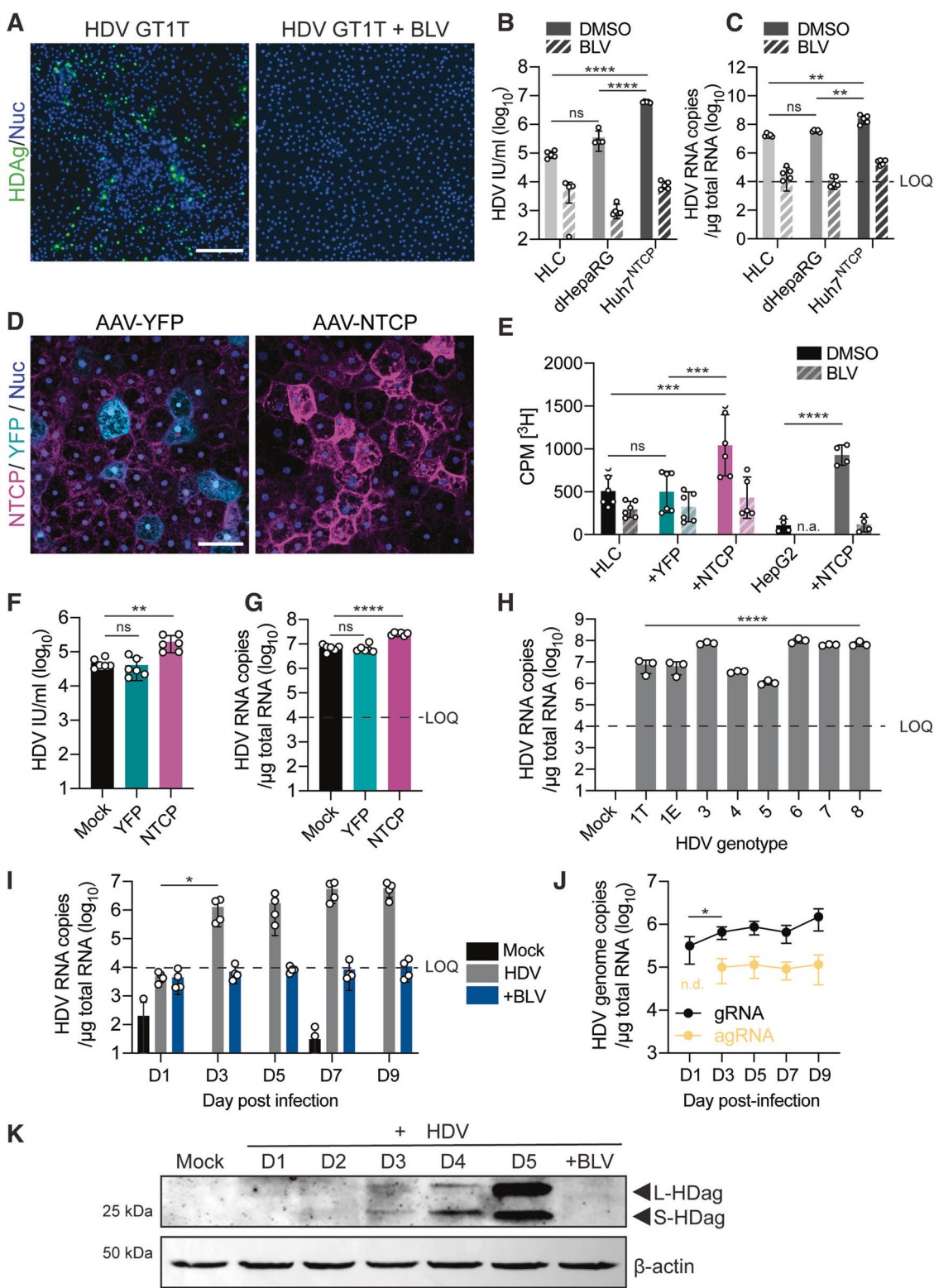

To assess the robustness of the system, we compared HDV infection of HLCs with conventional HDV culture systems. When using the same HDV inoculum, HLCs were similarly permissive to HDV infection (Fig. 1B) and supported HDV replication (Fig. 1C) as differentiated HepaRG cells. However, both HLCs and HepaRG

cells were much less permissive than carcinoma-derived Huh7 cells ectopically expressing human NTCP (Huh7$^{NTCP}$, Figs. 1B,C and EV1C).

We then investigated the possibility of further enhancing HDV infection by transducing HLCs with recombinant, NTCP-encoding

**Figure 1. Hepatocyte-like cells (HLCs) are susceptible to HDV infection in an NTCP-dependent manner.**

(A) HDV infection (MOI = 5 Int. Units/cell) of HLCs incubated with or without 500 nM entry inhibitor bulevirtide (BLV) was assessed by immunofluorescence staining (IF) against the HDV antigen (HDAg, green) 5 days post-infection (p.i.). GT: genotype. Scale bar = 200 μm. Images are representative of three independent HLC differentiations. (B, C) HLCs, differentiated (d)HepaRG, and Huh7$^{NTCP}$ cells were infected with HDV (MOI = 5 Int. Units/cell) with or without 500 nM BLV. HDV infection was analyzed by (B) counting HDAg-positive cells using CellProfiler or by (C) quantifying HDV genome copies by RT-qPCR. IU infectious unit. Dashed line: limit of quantification (LOQ). N = 5 biological replicates from two independent experiments. (D, E) HLCs were transduced with or without AAV6 (MOI = 10$^4$ viral genomes, vg/cell) encoding for YFP or NTCP two days before HDV infection. (D) NTCP was stained with Atto-MyrB-565 (magenta) two days post transduction. (E) [$^3$H]-taurocholate uptake was determined 2 days post transduction in HLCs ( + YFP or NTCP) or HepG2 cells ( + NTCP) treated or not with 1 μM BLV. CPM[$^3$H]: Tritium scintillation counts per minute. n.a: not assessed. N = 6 biological replicates from two independent HLC differentiations, N = 4 biological replicates (HepG2) from two independent experiments. (F, G) Two days post transduction, mature HLCs$^{YFP/NTCP}$ were infected with HDV (MOI = 5) and analyzed by (F) counting HDAg-positive cells or (G) quantifying HDV genome copies 5 days p.i. N = 6 biological replicates from two independent HLC differentiations. (H) HLCs were infected with the indicated HDV genotype (MOI = 15 Int. Units/cell for GTs 1 T, 1E, 4, 6, 7, 8; MOI = 30 Int. Units/cell for GTs 3 & 5) and 5 days p.i., HDV genome copies were quantified using RT-qPCR. MOIs used were based on different infectious titers of the genotypes obtained on Huh7$^{NTCP}$ cells to reach similar infection efficiency of HLCs. N = 3 biological replicates. (I–K) HDV infection (MOI = 5) of HLCs incubated with or without 500 nM BLV was analyzed over time by quantifying (I) HDV genome copies (N = 4 biological replicates from two independent HLC differentiations). (J) HDV genomic RNA (gRNA) and antigenomic RNA (agRNA) (N = 6 biological replicates from two independent HLC differentiations), as well as large (L-HDAg) and small (S-HDAg) HDV antigen expression by Western blot. n.d.: not detected. Data information: In (B, C, E–J) data are presented as mean ± SD. (B, C) Statistical analysis was performed by multiple comparisons of ordinary one-way ANOVA. Statistical significance in (B) was tested between HDV-infected DMSO-treated HLC and dHepaRG (P = 0.1384); between HDV-infected DMSO-treated HLC and Huh7$^{NTCP}$ (P < 0.0001) and between HDV-infected DMSO-treated dHepaRG and Huh7$^{NTCP}$ (P < 0.0001). Statistical significance in (C) was tested between HDV-infected DMSO-treated HLC and dHepaRG (P = 0.9402); between HDV-infected DMSO-treated HLC and Huh7$^{NTCP}$ (P = 0.0022) and between HDV-infected DMSO-treated dHepaRG and Huh7$^{NTCP}$ (P = 0.0037). Statistical significance in (E) was tested between DMSO-treated HLC and HLC$^{YFP}$ (P > 0.9999), between DMSO-treated HLC and HLC$^{NTCP}$ (P = 0.0003), between DMSO-treated HLC$^{YFP}$ and HLC$^{NTCP}$ (P = 0.0002) and between DMSO-treated HepG2 and HepG2$^{NTCP}$ (P < 0.0001) by multiple comparisons of two-way ANOVA. (F, G) Statistical analysis was performed by multiple comparisons of ordinary one-way ANOVA. Statistical significance in (F) was tested between HDV-infected HLC$^{Mock}$ and HLC$^{YFP}$ (P = 0.977) and between HDV-infected HLC$^{mock}$ and HLC$^{NTCP}$ (P = 0.0015). Statistical significance in (G) was tested between HDV-infected HLC$^{mock}$ and HLC$^{YFP}$ (P = 0.9713) and between HDV-infected HLC$^{Mock}$ and HLC$^{NTCP}$ (P < 0.0001). Statistical significance in (H) was tested among the HLC infected with different HDV genotypes (P < 0.0001) by ordinary one-way ANOVA. Statistical significance in (I) was tested between day 1- and day 3-harvested HDV-infected HLC (P = 0.0464) by an unpaired two-tailed t test. Statistical significance in (J) was tested between day 1- and day 3-harvested HDV-infected HLC (P = 0.0129) by an unpaired two-tailed t test. ****P < 0.0001; ***P < 0.001; **P < 0.01; *P < 0.05, n.s. non-significant. Source data are available online for this figure.

adeno-associated viruses (AAV) (Fig. 1D–G). Two days after AAV transduction, we confirmed endogenous and ectopic NTCP expression on the HLC surface membranes by microscopy using a fluorescently conjugated peptide that binds specifically to human NTCP (Fig. 1D), as well as by Western blot (WB) analysis (Fig. EV1D). We further confirmed the functionality of the ectopically expressed NTCP using a taurocholate transport assay (Lempp et al, 2019; Ni et al, 2014) (Fig. 1E). In contrast to hepatoma cells, in which we and others have observed a significant increase in HDV susceptibility upon NTCP replenishment (Yan et al, 2012), ectopic NTCP expression in HLCs enhanced HDV infection by only ~threefold (Fig. 1F,G). This suggests that, in contrast to hepatoma cells, endogenous NTCP expression levels are not a major rate-limiting factor for productive HDV infection of HLCs. Of note, we also used the hESC line RUES2 and the hiPSC line iPSC.C3A (Si-Tayeb et al, 2010) (a kind gift from Stephen Duncan, MUSC) to generate HLCs that were also permissive to HDV infection, demonstrating the reproducibility of the system.

Next, we infected HLCs with HDV GTs 3–8 and found that HLCs were permissive for all genotypes tested (Figs. 1H and EV1E), albeit with different infection efficiencies. Of note, since we previously found that the genotypes have different specific infectivities (Wang et al, 2021), we infected HLCs with adjusted MOIs to achieve similar infection rates.

We then analyzed the dynamics of HDV replication in HLCs by quantifying HDV genome copies over a period of 20 days post-infection. We observed a steep increase in HDV replication between days 1 and 3 post-infection (Fig. 1I), which subsequently plateaued. Starting from day 10 post-infection, HDV replication began to decline (Fig. EV1F), a pattern consistent with observation in other cell culture models, including primary human hepatocytes (PHH) (Gudima et al, 2007; Zhang et al, 2018). This decrease is

triggered by RNA editing, enabling HDV to transition from genome replication to packaging (Pérez-Vargas et al, 2021). In line with the dynamics of HDV replication in HLCs (Fig. 1I), we found a rapid accumulation of HDV genomes shortly after infection, with detectable HDV antigenomes appearing from day 3 post-infection onwards (Fig. 1J). This was followed by a shift from S-HDAg to L-HDAg production in HDV-infected HLCs around day 5 post-infection (Fig. 1K). Notably, although we could likewise detect antigenomic RNA in HDV-infected Huh7$^{NTCP}$ cells from day 3 post-infection on (Fig. EV1G), we detected an abundance of L-antigen only at later time points as compared to HLCs (Fig. EV1H). This difference in kinetics is interesting and should be followed up on in future studies. Importantly, we have demonstrated successful HDV genome editing in HLCs.

## Co-infection with HBV enables HDV to complete its life cycle in HLCs

Since HDV is a satellite virus of HBV, we next performed HBV genotype D and HDV GT1 co-infection of HLCs by inoculating both viruses at the same time (Fig. 2A) and counted the number of HLCs infected with either HBV (pink), HDV (green), or both viruses (white) (Fig. 2B). Although the majority of infected HLCs were single-positive for either HBV core (HBc) or HDV antigen (HDAg), we found that ~8% of the HBV-infected cells were double-positive for both HBc and HDAg, suggesting productive co-infection with both viruses (Fig. 2B).

To complement our analysis of HDV-related parameters in HDV mono-infected HLCs (Fig. 1), we also investigated HBV-related parameters in both HBV mono-infected and HBV/HDV co-infected HLCs. As shown in Fig. 2C, we successfully detected HBV covalently closed circular DNA (cccDNA), confirming robust HBV

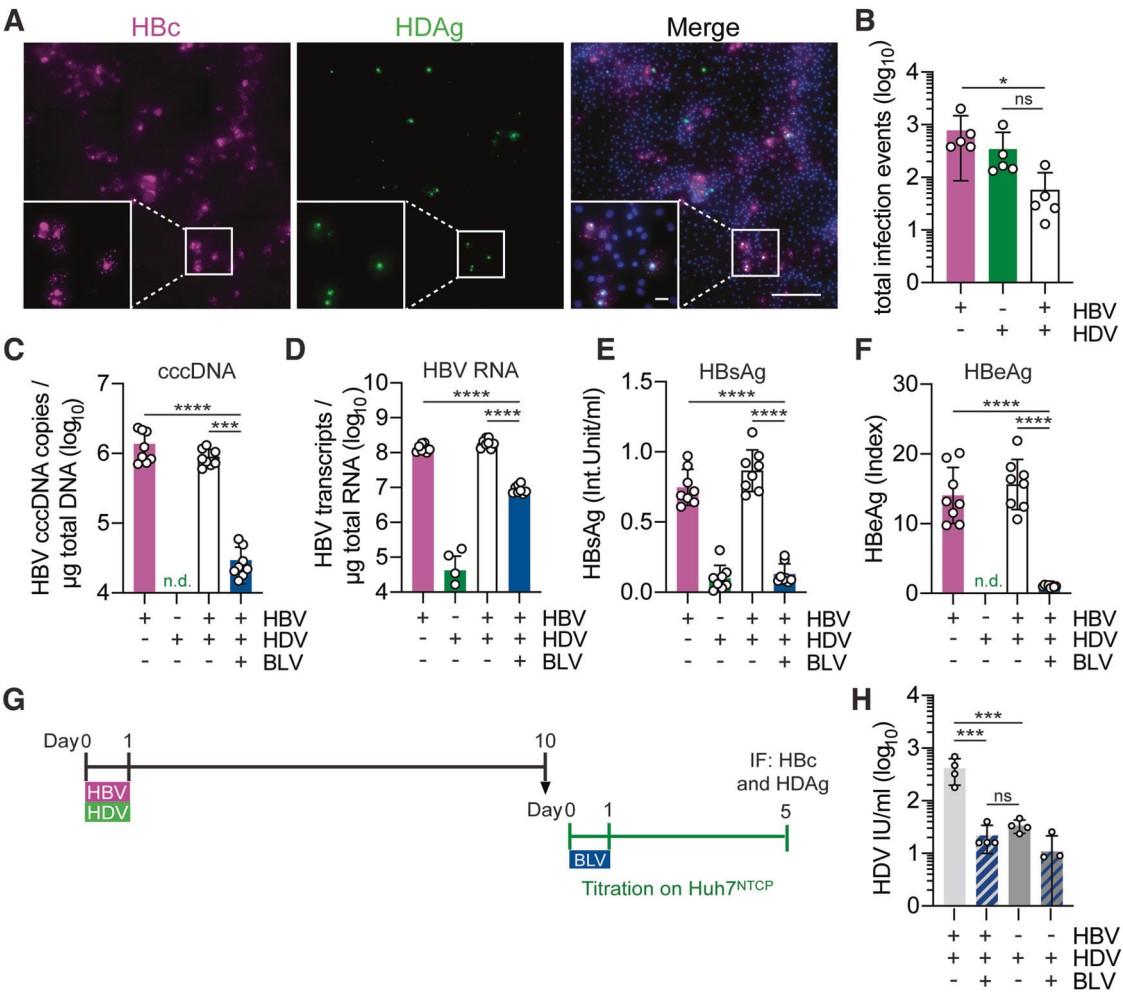

**Figure 2. Co-infection of with HBV enables HDV to complete its life cycle in HLCs.**

(A) HLCs were co-infected with HBV (MOI = 450 genome copies/cell) and HDV (MOI = 5 Int. Units/cell). Cells were stained against HBV core (HBc, magenta), HDAg (green), and nuclei (DAPI, blue) ten days p.i. Scale bar = 200 μm, scale bar of insets = 40 μm. Images are representative of three independent HLC differentiations. (B) The number of HBc-, HDAg- and double-positive cells were counted using ZEN imaging software to quantify HBV single (pink column), HDV single (green column) and co-infection events (white column) on 10 days p.i. $N = 5$ biological replicates from two independent HLC differentiations. (C–F) HLCs were infected with either HBV, HDV, or co-infected with HBV and HDV with or without 500 nm BLV and analyzed for (C) HBV cccDNA and (D) HBV total RNA in infected HLC lysates as well as for (E) HBsAg and (F) HBeAg secreted in the supernatant of infected HLCs (IU international units). n.d.: not detected. $N = 8$ biological replicates from two independent HLC differentiations. (G) The supernatant from HBV/HDV co- or HDV mono-infected HLCs collected on day ten p.i. was diluted 1:5 to infect Huh7$^{NTCP}$ cells with or without 500 nM BLV. (H) Infected Huh7$^{NTCP}$ cells were fixed, stained for HDAg, and analyzed using CellProfiler. $N = 4$ biological replicates from two independent experiments. Data information: In (B–F, H), data are presented as mean ± SD and statistical analysis was performed by multiple comparisons of ordinary one-way ANOVA. Statistical significance in (B) was tested between HBV mono- and HBV/HDV co-infected HLC ($P = 0.0493$) and between HDV mono- and HBV/HDV co-infected HLC ($P = 0.5276$). Statistical significance in (C) was tested between HBV mono- and BLV-treated HBV/HDV co-infected HLC ($P < 0.0001$) and between HBV/HDV co- and BLV-treated HBV/HDV co-infected HLC ($P = 0.0002$). Statistical significance in (D) was tested between HBV mono- and BLV-treated HBV/HDV co-infected HLC ($P < 0.0001$) and between HBV/HDV co- and BLV-treated HBV/HDV co-infected HLC ($P < 0.0001$). Statistical significance in (E) was tested between HBV mono- and BLV-treated HBV/HDV co-infected HLC ($P < 0.0001$) and between HBV/HDV co- and BLV-treated HBV/HDV co-infected HLC ($P < 0.0001$). Statistical significance in (F) was tested between HBV mono- and BLV-treated HBV/HDV co-infected HLC ($P < 0.0001$) and between HBV/HDV co- and BLV-treated HBV/HDV co-infected HLC ($P < 0.0001$). Statistical significance in (H) was tested between HBV/HDV co- and BLV-treated HBV/HDV co-infected HLC ($P = 0.0007$), between HBV/HDV co- and HDV mono-infected HLC ($P = 0.0009$) and between BLV-treated HBV/HDV co- and HDV mono-infected HLC ($P = 0.9975$). ****$P < 0.0001$; ***$P < 0.001$; *$P < 0.05$, n.s. non-significant. Source data are available online for this figure.

replication in these cells. We further observed comparable levels of total HBV RNA transcripts (Fig. 2D), as well as secretion of HBsAg (Fig. 2E) and HBV E-antigen (HBeAg, Fig. 2F) in the supernatant of both mono- and co-infected HLCs. These findings align with observations in other cell culture models, including primary PHH and HepG2$^{NTCP}$ cells (Tham et al, 2020).

Next, we investigated whether the co-infection with HBV enables HLCs to support the entire life cycle of HDV. To this end, we harvested infectious HDV progeny in the supernatant of HBV/HDV co-infected HLCs and titrated them on Huh7$^{NTCP}$ cells with or without BLV treatment (Fig. 2G). As shown in Fig. 2H, HDV progenies were only released from HLCs that were co-

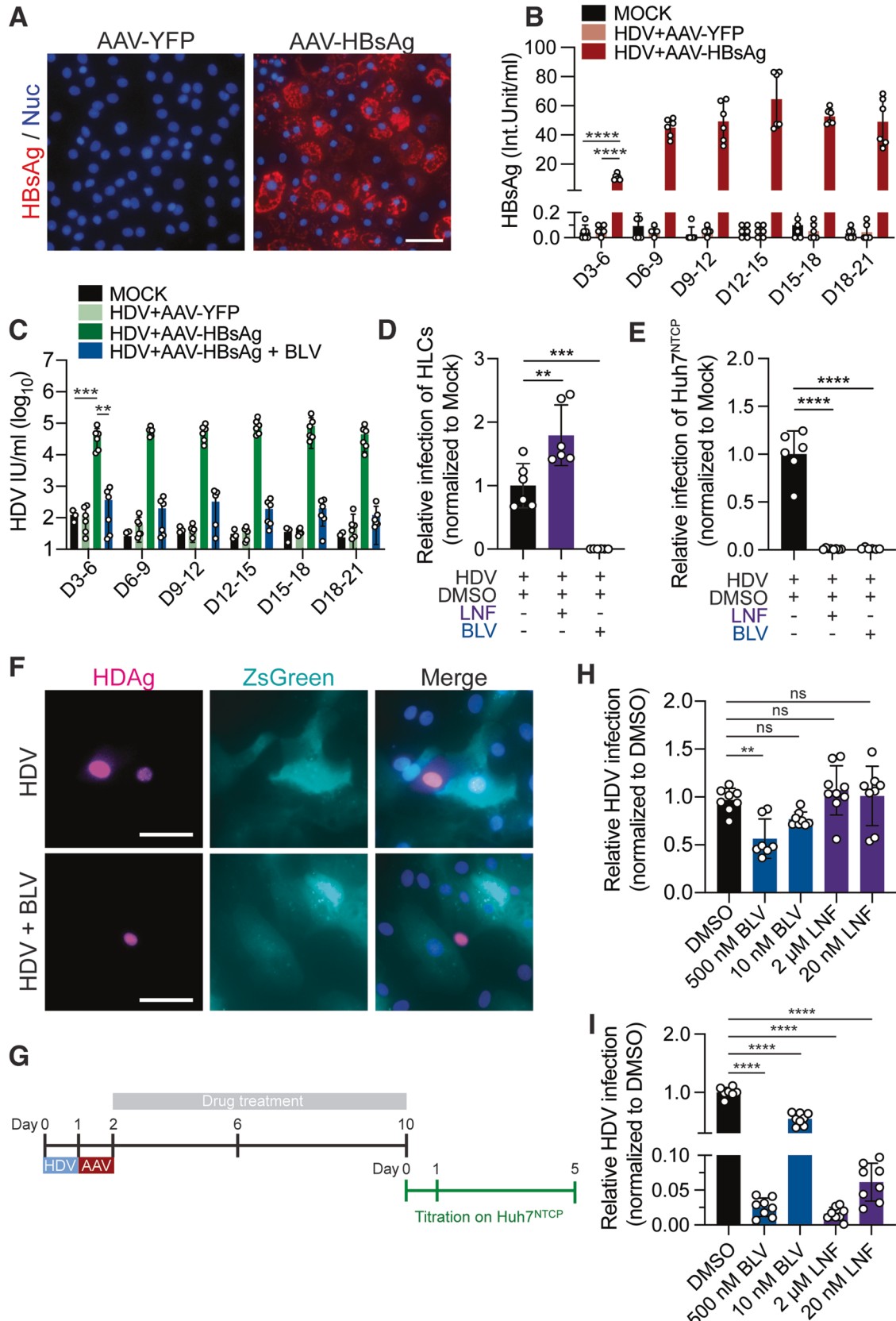

**Figure 3.   AAV transduction of HLCs with HBV surface antigens enables HDV extracellular spread and drug evaluation studies.**

(A) HLCs were infected with HDV (MOI = 5 Int. Units/cell) and the next day transduced with AAV6-YFP or AAV6-HBsAg. SN: supernatant. Nine days post transduction, HLCs were stained for HBsAg (red) and nuclei (blue). Images are representative of two independent differentiations. Scale bars = 50 μm. (B) HBsAg was quantified in the supernatant by ELISA collected at the end of indicated time periods. Int. Unit international unit. N = 6 biological replicates from two independent HLC differentiations. (C) Progeny HDV from HLCs harvested at the indicated time points was diluted 1:5 and used to infect Huh7$^{NTCP}$ cells with or without 500 nM BLV. Infected Huh7$^{NTCP}$ cells were fixed and stained for HDAg to quantify HDV infections. IU infectious unit. N = 6 biological replicates from 2 independent experiments. (D) HLCs were infected with HDV (MOI = 5 Int. Units/cell), transduced with AAV6-HBsAg and incubated with or without 500 nM BLV (between D0-D1 p.i.) or 2 μM Lonafarnib (LNF; between D0-D5 p.i.). HDV infection was quantified by counting HDAg-positive HLCs five days p.i. N = 6 biological replicates from two independent HLC differentiations. (E) The supernatant from these HLCs was diluted 1:5 to infect Huh7$^{NTCP}$ cells which were analyzed for HDV infection by HDAg staining five days p.i. N = biological replicates. N = 6 biological replicates from two independent HLC differentiations. (F) Wild-type HLCs were infected with HDV (MOI = 5 Int. Units/cell) and the next day transduced with AAV6-HBsAg. Two days post-infection, they were dissociated and co-cultured with ZsGreen expressing HLCs in the presence or absence of BLV. Eight days later, cells were fixed, stained, and imaged for HDAg (magenta), ZsGreen (cyan) and nuclei (DAPI, blue). Scale bar = 50 μm. (G) Experimental setup. HLCs were infected with HDV (MOI = 5 Int. Units/cell) and the next day transduced with AAV6-HBsAg. After removal of the inoculum on day 2 p.i., HLCs were incubated with drugs, which were replenished every four days. Ten days p.i., HLCs were fixed to analyze HDV infections and their culture supernatant was harvested for titration of HDV progenies on Huh7$^{NTCP}$ cells. (H) Relative HDV infection events normalized to vehicle DMSO-treated cells were quantified by counting HDAg-positive HLCs 10 days p.i. N = 8 biological replicates from three independent HLC differentiations. (I) The supernatant from HLCs was diluted 1:5 to infect Huh7$^{NTCP}$ cells, which were then analyzed for HDV infection by HDAg staining 5 days p.i. N = biological replicates. N = 8 biological replicates from three independent HLC differentiations. Data information: In (B–E, H, I) data are presented as mean ± SD and statistical analysis was performed by multiple comparisons of ordinary one-way ANOVA. Statistical significance in (B) was tested between HLC$^{Mock}$ and HLC$^{HBsAg}$ (days 3–6; $P < 0.0001$) and between HLC$^{YFP}$ and HLC$^{HBsAg}$ (days 3–6; $P < 0.0001$). Statistical significance in (C) was tested between HLC$^{YFP}$ and HLC$^{HBsAg}$ (days 3–6; $P = 0.0010$) and between HLC$^{HBsAg}$ and BLV-treated HLC$^{HBsAg}$ (days 3–6; $P = 0.0010$). Statistical significance in (D) was tested between DMSO- and LNF-treated HDV-infected HLC ($P = 0.0021$) and between DMSO- and BLV-treated HDV-infected HLC ($P = 0.0003$). Statistical significance in (E) was tested between DMSO- and LNF-treated HDV-infected HLC ($P < 0.0001$) and between DMSO- and BLV-treated HDV-infected HLC ($P < 0.0001$). Statistical significance in (H) was tested between DMSO- and 500 nM BLV-treated HDV-infected HLC ($P = 0.0071$), between DMSO- and 10 nM BLV-treated HDV-infected HLC ($P = 0.3508$), between DMSO- and 2 μM LNF-treated HDV-infected HLC ($P = 0.8643$) and between DMSO- and 20 nM LNF-treated HDV-infected HLC ($P = 0.9944$). Statistical significance in (I) was tested between DMSO- and 500 nM BLV-treated HDV-infected HLC ($P < 0.0001$), between DMSO- and 10 nM BLV-treated HDV-infected HLC ($P < 0.0001$), between DMSO- and 2 μM LNF-treated HDV-infected HLC ($P < 0.0001$) and between DMSO- and 20 nM LNF-treated HDV-infected HLC ($P < 0.0001$). ****$P < 0.0001$; ***$P < 0.001$; **$P < 0.01$, n.s. non-significant. Source data are available online for this figure.

infected with HBV/HDV but not mono-infected with HDV. The infection of HDV progenies could be fully blocked by the BLV treatment (Fig. 2H), demonstrating that HLCs can recapitulate the entire HDV life cycle.

## AAV transduction of HLCs with HBV surface antigens enables efficient HDV extracellular spread and drug evaluation studies

It has been shown experimentally that HBsAg derived from naturally integrated HBV DNA can mediate the production of infectious HDV virions in the absence of active HBV replication (Freitas et al, 2014; Tavanez et al, 2002). In support of this, high HDV viremia has been detected in HDV/HBV carriers with low or undetectable HBV levels (Pollicino et al, 2011), suggesting that HBV integrates are sufficient for expressing the HBV envelope proteins.

To mimic the expression of HBsAg from an alternative source than the cccDNA, we sought to use a previously identified AAV capsid that transduces HLCs with high efficiency (Zhang et al, 2022a) to express L-/M-/S-HBsAg in mature HLCs (HLCs$^{HBsAg}$, Fig. 3A). We replaced the original promoter in the AAV transgene vector with the authentic HBV promoter in order to obtain the optimal ratio of the three envelope proteins that allow particle formation, as we have shown previously (Lempp et al, 2019). Six days post-HDV infection, we detected secreted HBsAg in the supernatant of HLCs$^{HBsAg}$ (Fig. 3B), as well as HDV progeny capable of initiating secondary infections in Huh7$^{NTCP}$ cells (Fig. 3C). HDV-infected HLCs$^{HBsAg}$ continued to secrete both HBsAg and infectious progeny until the end of the observation period, i.e., up to 21 days post-HDV infection.

To ensure that the secondary infections were not carry-over events from the initial HDV inoculum, we used the HBV/HDV

entry inhibitor BLV and Lonafarnib (LNF), which prevents prenylation of the C-terminal Cys211 residue in L-HDAg and thus the envelopment of infectious HDV progeny (Lempp and Urban, 2017). LNF increased primary HDV infection of HLCs (Fig. 3D), in agreement with previous results (Lempp et al, 2019), but it fully blocked the assembly and release of infectious HDV progeny (Fig. 3E). Since BLV was already very effective in blocking primary HDV infection of HLCs (Fig. 3D), we did not observe any secondary infection in Huh7$^{NTCP}$ cells (Fig. 3E).

HDV extracellular spread, which occurs efficiently in vivo, is difficult to replicate in currently available in vitro models (Lempp et al, 2019; Verrier et al, 2016). Since HDV progeny can be released from HLCs$^{HBsAg}$, we next analyzed whether they also support HDV extracellular spread. The macromolecular crowding agent polyethylene glycol 8000 (PEG) is widely used to enhance HBV and HDV infection in cell culture (Le Seyec et al, 1999; Schulze et al, 2007; Verrier et al, 2016) and we likewise used PEG 8000 for our primary HDV infections of HLCs. To confirm that the extracellular spread of HDV in HLCs is possible without further addition of PEG, we validated that HLCs can be infected with HDV in the absence of PEG.

Next, we transduced WA09 cells at the pluripotent stem cell stage with lentiviruses to express ZsGreen and differentiated them into HLCs$^{ZsGreen}$. In parallel, we differentiated wild-type WA09 cells, infected them with HDV, and transduced them with AAVs to express HBsAg. We then detached both HLC types and mixed the "recipient" HLCs$^{ZsGreen}$ with "donor" HLCs$^{HDV/HBsAg}$. Eight days later, we analyzed the co-cultured donor and recipient HLCs for HDV infection by staining against HDAg. As shown in Fig. 3F, we found HDAg-positive HLCs$^{ZsGreen}$ in the vicinity of HDV-infected donor HLCs. The treatment with BLV completely prevented infections of HLCs$^{ZsGreen}$, showing unequivocally that the extracellular spread of HDV had occurred in our HLC culture.

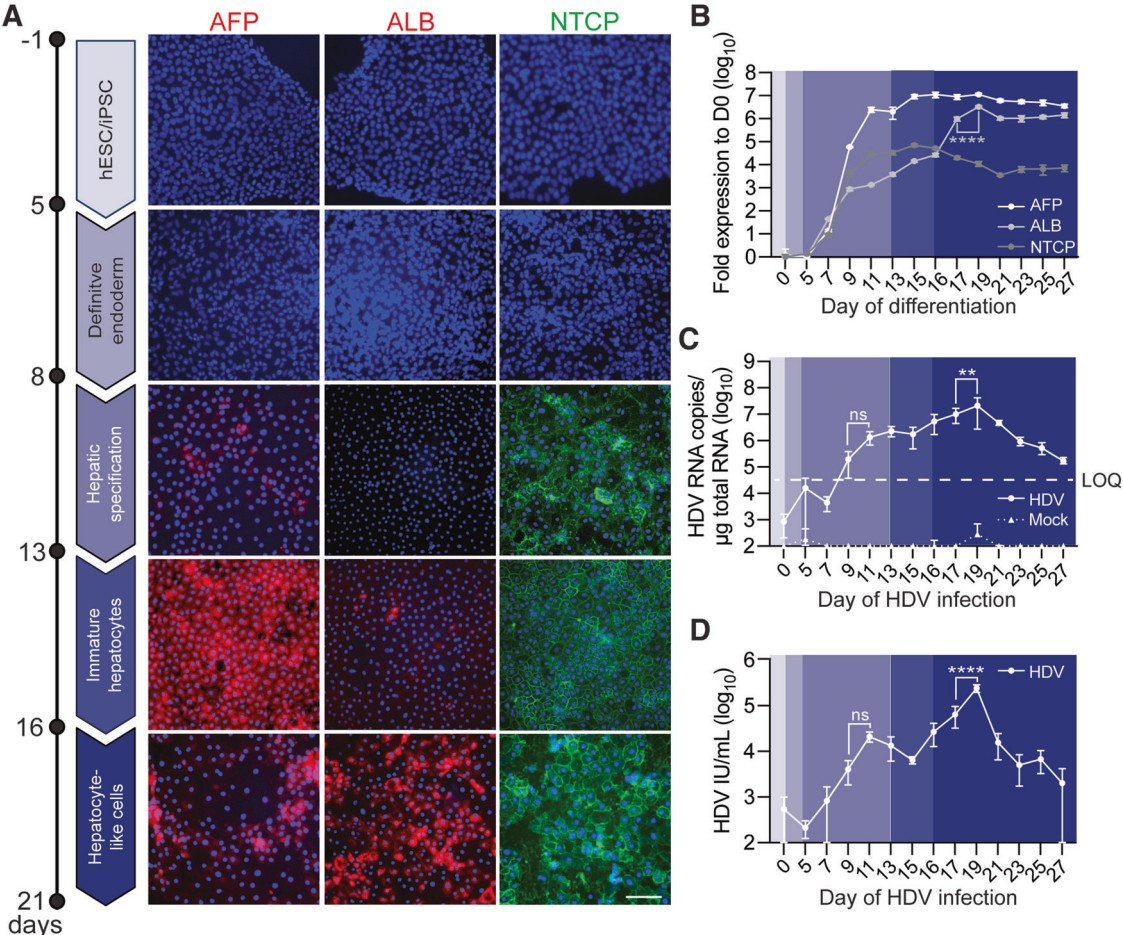

**Figure 4. HDV susceptibility along stem cell differentiation to hepatocyte-like cells.**

(A) Immunofluorescent images of cells stained against the nuclei (DAPI, blue) and with antibodies against alphafetoprotein (AFP, red), albumin (ALB, red) or Atto-MyrB[488] (NTCP, green) at the following stages during HLC differentiation: hPSCs, definitive endoderm, hepatic specification, immature, and mature hepatocyte-like cells. Images are representative of two independent HLC differentiations. Scale bar $= 100 \, \mu m$. (B) Cells were harvested for analyzing ALB, AFP, NTCP expression using RT-qPCR at the indicated day of the differentiation protocol. Results represent the mean ± SD of $N = 3$ biological replicates. (C, D) Cells were infected with HDV at the indicated day of the differentiation protocol and harvested five days p.i. HDV infection was analyzed by quantifying (C) HDV genome copies and (D) HDAg-positive cells using CellProfiler. Dashed line: LOQ. $N = 6$ biological replicates from two independent HLC differentiations. Data information: In (B–D), data are presented as mean ± SD and statistical analysis was performed by multiple comparisons of ordinary one-way ANOVA. Statistical significance in (B) was tested between day 17- and day 19-harvested hepatocytes (ALB; $P < 0.0001$). Statistical significance in (C) was tested between day 9- and day 11- HDV-infected hepatocytes ($P = 0.9995$) and between day 17- and day 19- HDV-infected hepatocytes ($P = 0.006$). Statistical significance in (D) was tested between day 9- and day 11- HDV-infected hepatocytes ($P = 0.4229$) and between day 17- and day 19- HDV-infected hepatocytes ($P < 0.0001$). ****$P < 0.0001$; **$P < 0.01$; n.s. non-significant. Source data are available online for this figure.

This prompted us to test the effect of available HDV antivirals on HDV spread in HLCs (Fig. 3G). First, we infected HLCs with HDV to allow primary infection. The next day, we transduced them with AAV-HBsAg. On the third day, we added the drugs to evaluate their impact on the secondary infections, and, accordingly, extracellular spread only. Since HLCs, like PHHs, do not proliferate (Levy et al, 2015), we could exclude cell division-mediated HDV spread.

Ten days post-infection, we observed that the treatment with a relatively high dose of 500 nM BLV reduced total HDV infection events by approximately 50% compared to untreated HLCs (Fig. 3H). In a dose-dependent manner, 10 nM BLV reduced the total number of HDV infections by only ~25%. Interestingly, treatment with LNF, which prevents HDV assembly, had no effect on the total number of HDV infections at day 10, but showed an

effect when the supernatant was used to titer infectious progeny on Huh7[NTCP] cells (Fig. 3I). This may be due to the observation that LNF enhances primary infections (Fig. 3D), in agreement with reports by others (Lange et al, 2023; Lempp et al, 2019). Of note, the high dose of BLV completely abrogated secondary infection of Huh7[NTCP] cells (Fig. 3I), although some progeny must have been released from primary infected HLCs. This suggests a carry-over of BLV in the inoculum and highlights the advantage of using a culture system that supports authentic and extracellular HDV spread.

## HDV susceptibility along HLC differentiation

Finally, we wanted to determine the stage along the differentiation process at which HLCs become susceptible to HDV infection. HLC

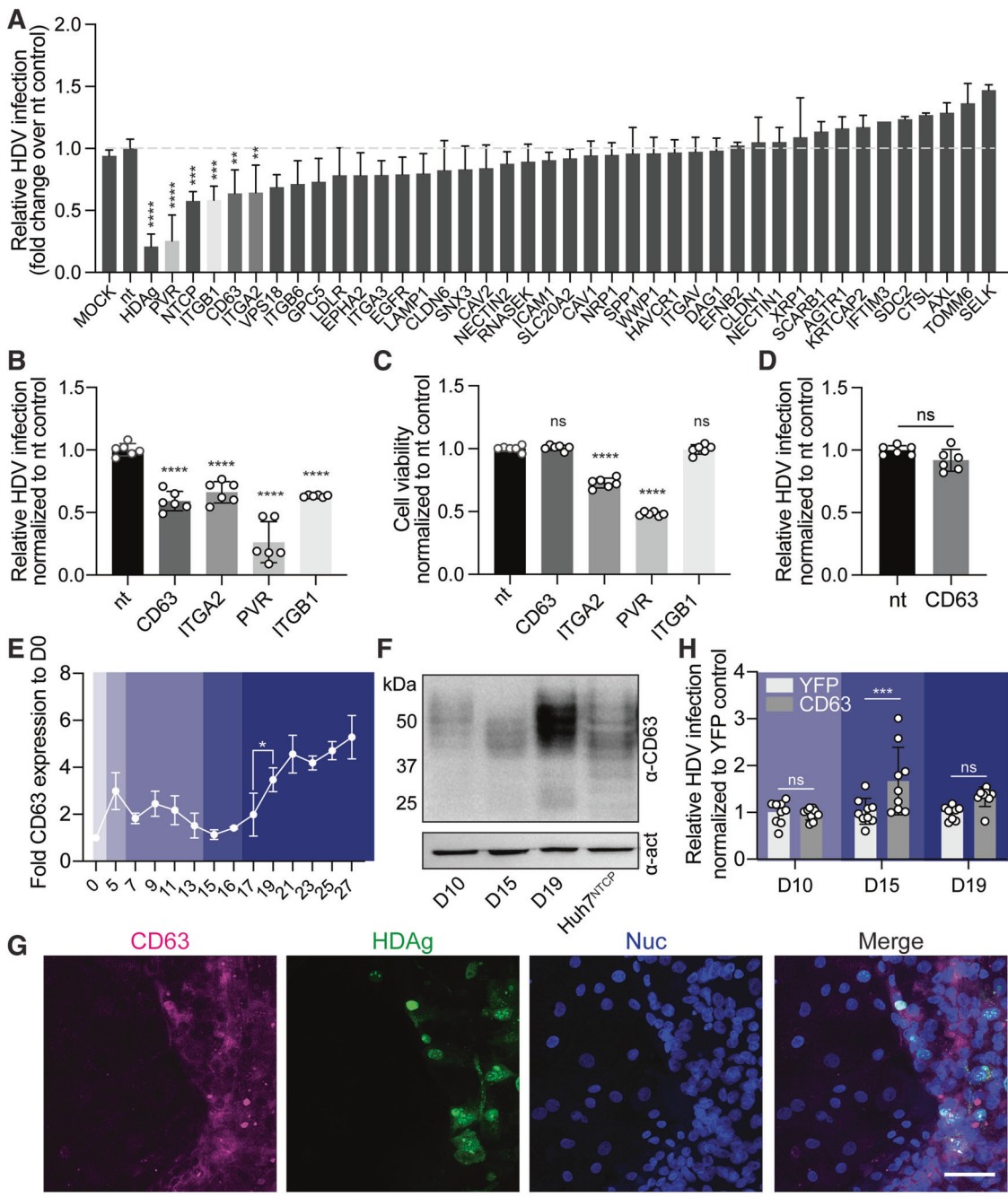

differentiation is based on a five-step protocol that mimics liver development, with each step involving the exposure of the cells to different culture media and growth factors: from hESC to definitive endoderm (DE), followed by hepatic specification, then immature, and finally mature hepatocytes (Fig. 4A). We monitored the progress of differentiation by analyzing markers of immature (alphafetoprotein, AFP) and mature hepatocytes (albumin/ALB and NTCP) at both protein (Fig. 4A) and transcript (Fig. 4B) levels. When we cultured DE cells in medium containing hepatocyte growth factor to induce hepatic specification, we observed a steep increase in AFP, ALB, and NTCP expression levels (Fig. 4B). While AFP levels reached a plateau by day 13, ALB expression did not

peak until day 19 in the final culture medium, when the cells reached their final stage of maturation. Interestingly, NTCP was already expressed on the cell surface during the hepatic specification stage, but decreased slightly during the final maturation step.

Then, we infected the cells with HDV either at the stem cell level (day 0), at the DE stage (day 5), or every other day during hepatocyte differentiation. At five days post-infection, we analyzed HDV infection by quantifying HDV RNA copy number (Fig. 4C) and HDAg-positive cells (Fig. 4D). Neither hESC nor DE cells were susceptible to HDV infection (Fig. 4C,D). Interestingly, when we bypassed the cell entry step by delivering the HDV genome via transfection, we found that hESCs were already capable of

◄ **Figure 5. siRNA screen reveals CD63 to be a potential co-factor of HDV cell entry which could be rate-limiting for infection of immature hepatocytes.**

(A) An siRNA screen of potential HDV host factors. Huh7$^{NTCP}$ cells were transfected with 50 nM on-target pool siRNAs directed against indicated genes and 72 h later infected with HDV (MOI = 1 Int. Units/cell). Relative HDV infection was normalized to non-target (nt) siRNA transfection and quantified by counting HDAg-positive cells five days p.i. N = 4 biological replicates from two independent experiments. (B, C) Four genes were selected and confirmed by separate siRNA transfection into Huh7$^{NTCP}$ cells and analyzed for (B) HDV infection after siRNA transfection and (C) cell toxicity. N = 6 biological replicates from two independent experiments. (D) Huh7$^{NTCP}$ cells were infected with HDV (MOI = 1 Int. Units/cell) and 24 h later transfected with nt- and CD63-siRNAs. HDV infections were quantified by counting HDAg-positive cells five days p.i. N = 6 biological replicates from three independent HLC differentiations. (E) HLCs were harvested for analyzing CD63 expression using RT-qPCR at the indicated day of the differentiation protocol. N = 3 biological replicates. (F) Western blot analysis of Huh7$^{NTCP}$ cells as control and HLC cell lysates harvested at indicated day of the differentiation protocol, for CD63 and β-actin (act) expression using respective antibodies. (G) Mature HLCs were infected with HDV, fixed 5 days post-infection, and stained for CD63 (magenta), HDAg (green), and the nucleus (blue). Images were taken on the Airyscan confocal microscope. Shown are ×40 tile image stacks of maximum projections with the two HLC populations. Scale bar = 50 μm. (H) Cells at indicated day of the differentiation protocol were transduced with AAV6-YFP or AAV6-CD63 and two days later, infected with HDV. Relative HDV infections were quantified by counting HDAg-positive cells five days p.i. N = 9 biological replicates from three independent HLC differentiations. Data information: In (A–E, H) data are presented as mean ± SD. In (A–C, E) statistical analysis was performed by multiple comparisons of ordinary one-way ANOVA. Statistical significance in (A) was tested by comparing each sample against the non-target knockdown control ($P_{HDAg}$ < 0.0001; $P_{PVR}$ < 0.0001; $P_{NTCP}$ = 0.0007; $P_{ITGB1}$ = 0.0008; $P_{CD63}$ = 0.0036; $P_{ITGA2}$ = 0.0042). Statistical significance in (B) was tested by comparing each sample against the non-target knockdown control ($P_{CD63}$ < 0.0001; $P_{ITGA2}$ < 0.0001; $P_{PVR}$ < 0.0001; $P_{ITGB1}$ < 0.0001). Statistical significance in (C) was tested by comparing each sample against the non-target knockdown control ($P_{CD63}$ = 0.9908; $P_{ITGA2}$ < 0.0001; $P_{PVR}$ < 0.0001; $P_{ITGB1}$ = 0.9908). Statistical significance in (D) was tested between HDV-infected non-target and CD63 knockdown Huh7$^{NTCP}$ (P = 0.0657) by unpaired two-tailed t test. Statistical significance in (E) was tested between day 17- and day 19-harvested hepatocytes (P = 0.0303). Statistical significance in (H) was tested between HDV-infected hepatocyte$^{YFP}$ and hepatocyte$^{CD63}$ at different time points, respectively, by multiple comparisons of two-way ANOVA ($P_{day10}$ = 0.9962; $P_{day15}$ = 0.0008; $P_{day19}$ = 0.1345). ****P < 0.0001; **P < 0.01; n.s. non-significant. Source data are available online for this figure.

replicating HDV, as evidenced by the detection of L-HDAg in transfected cells (Fig. EV2A).

We found that hepatic progenitors became susceptible to HDV infection as early as day 11 after the start of the differentiation protocol, which is likely to be controlled by the expression of NTCP. Interestingly, at day 15, immature hepatocytes were less susceptible to HDV infection than progenitors, even though they expressed NTCP on their surface. We then observed a second peak of HDV infection as the cells acquired a fully mature hepatocyte profile, as evidenced by the peak of ALB expression at day 19 (Fig. 4B). Remarkably, NTCP expression was not further enhanced at this time point, suggesting that other factors were limiting HDV in the immature state. To identify the exact day, we further refined and infected the cells with HDV every day between days 17 and 19 (Fig EV2B,C) of the differentiation protocol. We identified day 18 as the differentiation day when cells were the most susceptible to HDV infection, approximately 2 days after switching to the final maturation medium. From day 19 onwards, mature HLCs became less susceptible to HDV infection, probably due to a further decrease in NTCP expression (Fig. 4B).

The increased permissiveness observed in mature HLCs at day 18 could be due to the upregulation of a host co-factor or the downregulation of a host restriction factor for HDV infection. Therefore, we compared the transcriptome profile of the cells between day 17 and day 18 (Fig. EV3). Whole-transcriptome expression profiling and gene ontology (GO) term enrichment analysis revealed up- or downregulation of genes enriched in many pathways, such as liver development and regeneration (Fig. EV3A). Consistently, individual markers of mature hepatocytes were upregulated, whereas markers of immature hepatocytes were downregulated by day 18 (Fig. EV3B).

Importantly, the GO analysis revealed the upregulation of many genes that have been described to be involved in the entry process of viruses, including HBV (Fig. EV3C). In particular, genes encoding proteins that were found to interact with NTCP (Palatini et al, 2022), as well as described HBV and HDV co-entry factors, including recently identified Neuropilin 1 (Yu et al, 2024) were upregulated in mature HLCs at day 18. We validated the

upregulation of selected entry cofactors along HLC differentiation by Western blot analysis (Fig. EV3D).

## CD63 is a potential co-factor of HDV entry and could be rate-limiting for infection of immature hepatocytes

To identify a potential host factor that was expressed in mature HLCs and responsible for enhanced HDV infection, we performed a small siRNA screen in Huh7$^{NTCP}$ cells. We selected siRNAs targeting genes encoding membrane receptors and viral entry cofactors that were upregulated in mature HLCs as revealed by transcriptome analysis (Fig. EV3C). As positive controls, we also used siRNAs targeting NTCP and the HDAg coding region in the HDV RNA genome.

As shown in Fig. 5A, the downregulation of several genes reduced HDV infection by up to 80% compared to the transfection with a non-target (nt) control siRNA. We then selected the top four candidate genes for further validation (Fig. 5B): the poliovirus receptor (PVR), CD63, as well as integrins beta 1 (ITGβ1) and alpha 2 (ITGα2), which can form a functional ITGβ1/α2 heterodimer. We confirmed the downregulation of these genes by Western blot analysis (Fig. EV4A) and their impact on HDV infection of Huh7$^{NTCP}$ cells (Fig. 5B). However, the downregulation of PVR and ITGA2 had a pronounced effect on cell attachment and/or cell viability (Fig. 5C). Since the downregulation of CD63 reduced HDV infection by up to 50% without affecting NTCP or EGFR surface expression (Fig. EV4B) or cell viability (Fig. 5C), we decided to follow up on this factor.

To investigate whether CD63 plays a role either in the early steps of the HDV life cycle, including cell entry, or in later steps such as genome replication, we also delivered the CD63-targeting siRNA one day after HDV infection of Huh7$^{NTCP}$ cells (Fig. 5D). Only when CD63 was downregulated before (Fig. 5A,B), but not after HDV infection, we observed a significant reduction, suggesting that CD63 plays a role in the early steps of HDV infection.

When analyzing CD63 expression along HLC differentiation, we found a distinct upregulation in mature HLCs as compared to immature HLCs, at both transcript and protein levels (Fig. 5E,F).

Interestingly, we also found that CD63 appeared to be less glycosylated in immature hepatocytes as compared to mature hepatocytes (Fig. 5F), mirroring observations made by others in immature and mature dendritic cells (Engering et al, 2003). As described above and as shown in Fig. 1A and Fig EV1A, we observed throughout our experiments that HDV appeared to preferentially infect the highly confluent HLC population. Indeed, by confocal microscopy analysis, we found that CD63 was more expressed in the highly confluent and HDV-permissive HLC population compared to the less confluent and HDV-non-permissive HLC population (Figs. 5G and EV4C). Although the highly confluent HLC population appeared to be less mature (Fig. EV5A), they expressed higher levels of previously reported HBV/HDV cell entry factors such as LAMP1, SCARB1, and EGFR (Fig. EV5B). Therefore, we propose that CD63 together with these factors may render this HLC subpopulation more susceptible to HDV infection.

We then wanted to rescue CD63 expression at different stages of hepatocyte maturation. We transduced cells with AAV6 on days 8, 13, and 17 of the differentiation protocol, verified ectopic CD63 expression at the different stages (Fig. EV4D) and infected them with HDV 2 days later at each time point (Fig. 5H). Ectopic CD63 expression resulted in a greater, up to threefold increase in HDV infection of immature hepatocytes, but only a 1.2-fold increase in mature hepatocytes, demonstrating that CD63 may be a rate-limiting factor for HDV infection of immature hepatocytes. Future studies will provide a more mechanistic understanding of how CD63 may be involved in HDV entry. Here, we have provided guidance on how stem cell differentiation culture systems can be used as a platform to identify novel cofactors for viral infection.

# Discussion

Historically, studies of human hepatotropic viruses have been hampered by the lack of physiologically relevant liver cell culture models. Similarly, for HDV studies, the lack of reproducible cell culture systems that mimic the physiological state of hepatocytes in vivo has hampered our progress in understanding important aspects of HDV biology. Previous work by us and others have demonstrated the advantages of hPSC-derived HLCs for HBV, HCV, and HEV investigations (Carpentier et al, 2020; Wu et al, 2018; Xia et al, 2019). However, although they share the same tissue tropism, these viruses differ in their biology and various aspects of their individual life cycles remain to be elucidated. Here we have demonstrated the applicability of HLCs for HDV studies.

## HLCs to study the entire life cycle of HDV and HBV/HDV co-infections

We found that HLCs were readily permissive for HDV and HBV infection and that the ectopic NTCP expression under a foreign promoter did not dramatically increase HDV infection. These results are in stark contrast to hepatoma cells (Yan et al, 2012) and highlight the authenticity of HLCs. Thus, HLCs allow studies of HBV and HDV infection under the physiological regulation of NTCP expression, which may depend on bile acid concentrations, inflammatory cytokines, and other factors (reviewed in (Appelman et al, 2021)).

Interestingly, although the overall kinetics of HDV genome replication were similar in HLCs compared to other cell culture models such as Huh7$^{NTCP}$ cells, we detected an earlier switch to progeny assembly in HLCs. Although we did not explore the underlying causes in this study, future work should aim to identify these differences, e.g., by comparing ADAR1-mediated antigenome editing, which may be responsible for abundant L-HDAg production at an earlier stage of infection.

During the preparation on this manuscript, Lange and colleagues (Lange et al, 2023) used HLCs to study the innate immune response to HDV mono-infection. Here, in contrast, we studied HDV biology in the context of its helper virus. By either co-infecting with HBV or by genetically engineering HLCs to express HBsAg, we detected infectious HDV progeny in the HLC culture medium, demonstrating that HLCs can recapitulate the entire HDV life cycle and even extracellular HDV spread.

Both de novo infection and cell division-mediated spread contribute to HDV persistence in patients with CHD (Zhang and Urban, 2021), but to our knowledge, cell-to-cell spread as a mode of HDV transmission has not been demonstrated yet. We have previously shown that HDV spreads in hepatoma cells upon cell division and that it is highly dependent on the innate immune competence of the respective cell line used (Zhang et al, 2022b). HLCs, similar to other primary hepatocyte culture models like PHH and differentiated HepaRG cells, no longer proliferate, and thus do not support spread through cell division.

Extracellular spread of HDV is difficult to replicate in available in vitro models. On the one hand, rapid dedifferentiation and rapid decrease of NTCP expression after seeding severely limit the use of PHHs for HDV spread studies (Heuschkel et al, 2021). On the other hand, hepatoma cells ectopically expressing NTCP and HBsAg only poorly support extracellular spread, as shown in our recent study (Lempp et al, 2019). Here, we have shown that HLCs support extracellular HDV spread and thus provide a physiologically relevant, reproducible, and non-proliferative cell model to study the underlying determinants.

The genetic manipulation of HLCs via AAVs allowed us to mimic the expression of HBsAg from integrates rather than cccDNA, which has been reported to occur in HBV patients receiving nucleos(t)ide analog (Nuc) treatment (Ringlander et al, 2020) and highlights the potential of genetic manipulation of HLCs as a powerful tool to study the molecular mechanisms underlying HDV infection. Although there is evidence of integration, AAV genomes remain predominantly episomal. Importantly, we used them in this study only to investigate HDV progeny assembly and release outside the context of active HBV infection, not to mimic HBV integrates. In the future, genetic manipulation will allow the introduction of single-nucleotide polymorphisms identified in genome-wide association studies.

## HBsAg-HLCs for anti-HDV treatment evaluation

HLCs express members of the CYP450 family and their drug response correlates with that of PHHs (Takayama et al, 2014). Therefore, numerous previous studies have proposed HLCs as a platform for drug toxicity (Greenhough et al, 2010) and drug evaluation studies (Szkolnicka et al, 2014). In addition, owing to the self-renewal of stem cells, HLCs can be easily scaled up and used for high-throughput drug screening (Cayo et al, 2017). Finally, patient-

specific iPSCs can be generated from clinically relevant individuals to create personalized disease and drug evaluation models.

Since HLCs$^{HBsAg}$ support extracellular HDV spread, we used them to test the currently available drugs specifically developed to treat HDV infection. First, we confirmed that BLV completely blocked HDV entry into HLCs. In contrast, the application of LNF led to more detectable primary HDV infections, which may be related to the intra-hepatocellular accumulation of non-farnesylated and therefore non-inhibitory L-HDAg (Lempp et al, 2019). As a result, we did not observe a significant change in total HDV infection after treatment with LNF in our HLC spread assay. Only when we titrated HDV progeny separately and analyzed secondary infections, we could observe the effect of LNF. Our results highlight the advantage of studying drug efficacy in a system that faithfully recapitulates spread, especially since currently available and specific anti-HDV regimens target virus entry and assembly.

## Challenging HLCs along their differentiation revealed CD63 as a potential HDV/HBV entry factor

To our surprise, NTCP was expressed very early during HLC differentiation which mimics liver development. This early expression of NTCP rendered hepatic progenitors susceptible to HDV infection, consistent with previous studies showing that non-hepatic cell lines become susceptible to HDV infection upon ectopic NTCP expression (Yan et al, 2013). In agreement, we found that stem cells were already capable of replicating the HDV genome when delivered by transfection, bypassing the entry step, consistent with the findings of Lange et al (Lange et al, 2023) .

Surprisingly, we discovered that fully mature HLCs at day 18 of differentiation were more susceptible to HDV infection than less mature cells. This suggested the presence of one or more hepatocyte-specific cofactors that facilitate HDV entry or replication. We excluded NTCP as the responsible determinant because its expression was slightly decreased in fully mature hepatocytes. Our subsequent analysis using a small siRNA screen targeting selected genes from the transcriptome analysis, revealed CD63 as potential HDV entry factors. CD63 appeared to be rate-limiting for HDV infection of immature hepatocytes, which was alleviated by ectopic expression of CD63. While CD63 is best known for its role in exosomal egress and has been reported to be involved in HBV assembly and egress (Ninomiya et al, 2021), it is also a critical component of late endosomes and facilitates vesicular trafficking through endosomal pathways (Pols and Klumperman, 2009). As such, it has been shown to be involved in the entry of other viruses (Raaben et al, 2017; Zona et al, 2014). Therefore, we speculate that similar to what has been described for Lujo virus (Tominaga et al, 2014), CD63 may be involved in either HDV trafficking and/or virus fusion in the late endosome or lysosome, determining productive cell entry.

It is important to note that CD63 is likely not the only factor governing enhanced infection of mature and highly dense HLCs, and future research should aim to identify and confirm such other factors. In addition, mechanistic studies will be needed to confirm and elucidate the role of CD63 in HBV/HDV entry. In this study, we demonstrated how challenging cells along the stem cell differentiation process can be a dynamic platform for discovering new host factors for virus infection.

## Limitations of using HLCs for HBV/HDV infection studies

While we have shown that HLCs support HBV and HDV infection, their susceptibility remained inferior to that of hepatoma cells overexpressing NTCP, possibly due to their immature nature. In addition, as compared to other hepatocellular culture models, we have observed an induction of L-HDAg at similar levels to S-HDAg at the early stages of HDV infection in HLCs, resulting in a rapid inhibition of HDV replication. The underlying causes of this early switch should be addressed in future studies, with the potential to identify a host factor that could repress HDV replication efficiently.

Finally, we observed that HDV preferentially infected the highly confluent HLC population, but not the less confluent HLC population, although both populations expressed ALB and NTCP at similar levels. While we found a potential correlative expression with CD63, future studies will identify other critical cofactors for HDV infection that are absent or potential limiting factors that are present in the different HLC subpopulations. Comparing the genetic landscape between the two populations could potentially lead to the identification of such factors. These factors could then be genetically modified by lentiviral or AAV transduction, as we have done in this study, to generate highly permissive HLCs.

In general, stem cell culture and differentiation remain expensive and time-consuming. Similar to other primary hepatocyte culture models, even though the HLC system could be cultured for at least three weeks, they unavoidably deteriorate over time (Verrier et al, 2016). In the absence of a deep understanding of the molecular mechanisms of liver development and regeneration (Miyajima et al, 2014; Tanimizu and Miyajima, 2007), a wide range of HLC differentiation protocols have been published for different applications. A better understanding of liver development is needed and should lead to the development of more robust and potentially commercially available HLC differentiation kits in the future. This should make the system available to all researchers in the field with the overarching goal of advancing molecular HDV studies and developing curative and alternative therapies for chronic HDV patients.

# Methods

**Reagents and tools table**

| Reagent/resource | Reference or source | Identifier or catalog number |
|---|---|---|
| **Experimental models** | | |
| WA09 (Wicell) (*H. sapiens*) | WiCell (Thomson et al, 1998) | WAe009-A |
| HepaRG (*H. sapiens*) | Gripon et al, 2002 | CVCL_9720 |
| Huh7$^{NTCP}$ (*H. sapiens*) | Ni et al, 2014 | N/A |
| HepG2 (*H. sapiens*) | Ni et al, 2014 | CVCL_0027 |
| HepG2$^{NTCP}$ (*H. sapiens*) | Ni et al, 2014 | N/A |
| HEK-293 (*H. sapiens*) | Maurer et al, 2022 | CRL-1573 |
| **Recombinant DNA** | | |
| pJC126 | Gudima et al, 2002 | N/A |
| pscAAV-CMV-EYFP-BGHpolyA | Börner et al, 2020 | N/A |

| Reagent/resource | Reference or source | Identifier or catalog number |
|---|---|---|
| pSSV9-AAV-CMV-EYFP-BGHpolyA | Wolff et al, 2019 | N/A |
| pscAAV-CMV-hNTCP-BGHpolyA | This study | N/A |
| pscAAV-CMV-CD63-BGHpolyA | This study | N/A |
| pSSV9-AAV-CMV-HBsAg-BGHpolyA | This study | N/A |
| **Antibodies** | | |
| Monoclonal mouse/human anti-HDAg (1:3000 for immunofluorescence) | Wang et al, 2021 | FD3A7 |
| Rabbit anti-HBcAg (1:1000 for immunofluorescence) | Ni et al, 2014 | H363 |
| Rabbit anti-L-HDAg (1:5000 for immunofluorescence) | Lempp et al, 2019 | N/A |
| Human anti-HBsAg (1:1000 for immunofluorescence) | Davide Corti, Humabs BioMed | HBD87 |
| Rabbit anti-FoxA2 (1:400 for immunofluorescence) | Cell Signaling | 8186 |
| Mouse anti-AFP (1:1000 for immunofluorescence) | Sigma-Aldrich | A8452 |
| Mouse anti-ALB (1:1000 for immunofluorescence) | Cedarlane | CL2513A |
| Mouse anti-CD63 (1:400 for immunofluorescence) | Santa Cruz Biotechnology | sc-5275 |
| Rabbit anti-ADAR1 (1:1000 for immunofluorescence) | Cell Signaling | 14175 |
| Rabbit anti-LAMP1 (1:200 for immunofluorescence; 1:1000 for western blot) | Cell Signaling | 9091 |
| Rabbit anti-SCARB1 (1:500 for immunofluorescence; 1:1000 for western blot) | Novus Biologicals | NB400-104 |
| Rabbit anti-EGFR (1:00 for immunofluorescence; 1:1000 for western blot) | Cell signaling | 4267 |
| Rabbit anti-LDLR (1:1000 for western blot) | Abcam | ab30532 |
| Mouse anti-CD63 (1:100 for western blot) | Invitrogen | 10628D |
| Rabbit anti-PVR (1:1000 for western blot) | Sigma-Aldrich | ZRB1036 |
| Mouse anti-ITGβ1 (1:1000 for western blot) | Santa Cruz | sc-53711 |
| Rabbit anti-ITGα2 (1:100 for western blot) | Abcam | ab181548 |
| Mouse anti-β-actin (1:4000 for western blot) | Sigma-Aldrich | A2228 |
| HPR-coupled anti-mouse (1:4000 for western blot) | Jackson ImmunoResearch, Ely, UK | AB_2307392 |
| HPR-coupled anti-rabbit (1:4000 for western blot) | Jackson ImmunoResearch, Ely, UK | AB_2307392 |
| **Oligonucleotides and other sequence-based reagents** | | |
| PCR primers | This study | Appendix Table S1 |

| Reagent/resource | Reference or source | Identifier or catalog number |
|---|---|---|
| **Chemicals, enzymes and other reagents** | | |
| DMEM, high glucose, GlutaMAX™ | Gibco | 61965 |
| Fetal Bovine Serum (FBS) | Capricorn Scientific | FBS-HI-11A |
| Penicillin–Streptomycin | Gibco | 15140-122 |
| Matrigel® Matrix Basement Membrane Growth Factor Reduced (GFR) | Corning | 354230 |
| ReLeSR™ | Stem Cell Technologies | 05872 |
| mTeSR™1 | Stem Cell Technologies | 85850 |
| STEMdiff™ Definitive Endoderm Kit | Stem Cell Technologies | 05110 |
| DPBS (1X) | Gibco | 14190-144 |
| Rock inhibitor | Stemcell Technologies | Y-27632 |
| Accutase Cell Detachment Solution | Pan Biotech | P10-21500 |
| KnockOut™ DMEM/F-12 (1X) | Gibco | 12660-012 |
| KNOCKOUT™ SR, Serum Replacement for ES cells | Gibco | 10828-028 |
| Dimethylsulfoxide (DMSO) | Sigma-Aldrich | D2650 |
| MEM NEAA (100X) | Gibco | 11140-050 |
| GlutaMAXTM-1 100X) | Gibco | 35050-061 |
| Human HGF | PeproTech | 100-39 |
| Dexamethasone | Sigma | D4902 |
| HCM™ Hepatocyte Culture Medium BulletKit™ | Lonza | CC-3198 |
| rhOncostatin M recombinant human (OSM) | RD system | 295-DM |
| polyethylene glycol 8000 | Sigma-Aldrich | 89510 |
| Dimethylsulfoxide (DMSO) | Roth | A994.2 |
| OptiMEM™ I Reduced Serum Medium | Gibco | 31985047 |
| Lipofectamine™ Stem Transfection Reagent | Life Technologies | STEM00003 |
| paraformaldehyde | Science Services | 15710 |
| NucleoSpin RNA kit | Macherey-Nagel | 740955 |
| High-capacity cDNA reverse-transcription kit | Applied Biosystems | 4368814 |
| Luna® Universal Probe qPCR Master Mix | New England Biolabs | M3004 |
| iTaq™ Universal SYBR ® Green Supermix | Bio-rad | 1725125 |
| NEBNext Ultra II Directional RNA Preparation Kit | New England Biolabs | E7760 |
| siRNAs | Dharmacon | N/A |
| Lipofectamine™ RNAiMAX Transfection Reagent | Life Technologies | 13778075 |
| Pierce™ RIPA-Buffer | Life Technologies | 89900 |
| cOmplete™, Mini Protease Inhibitor Cocktail | Roche | 11836153001 |
| PNGaseF | New England Biolabs | P0704 |

| Reagent/resource | Reference or source | Identifier or catalog number |
|---|---|---|
| RNase H | New England Biolabs | M0297S |
| RNase inhibitors | Applied Biosystems | N8080119 |
| SuperScript™ IV Reverse Transcriptase | Invitrogen | 18090050 |
| DNA Clean & Concentrator-5 | Zymo Research | D4013 |
| Quick-DNA/RNA Miniprep Plus Kit | Zymo Research | D7003 |
| T5 Exonuclease | New England Biolabs | M0363 |
| Lonafarnib | Absource Diagnostic | S2797-0005 |
| Bulevirtide | Bachem | N/A |
| **Software** | | |
| GraphPad PRISM.8 | GraphPad software | N/A |
| Adobe Illustrator 2024 | Adobe | N/A |
| Salmon version 0.15.2 | Patro et al, 2017 | N/A |
| tximport version 1.24.0 | Soneson et al, 2015 | N/A |
| DESeq2 version 1.36.0 | Love et al, 2014 | N/A |
| topGO version 2.48.0 | Alexa and Rahnenfuhrer, 2016 | N/A |
| Fiji/ImageJ | National Institutes of Health | N/A |
| ZEN blue edition 3.4 | Carl Zeiss AG | N/A |
| CellProfiler | https://cellprofiler.org/ | N/A |
| BioRender | https://www.biorender.com/ | N/A |
| **Other** | | |
| Zeiss Cell Discoverer 7 (CD7) microscope | Carl Zeiss AG | N/A |
| Enhanced resolution microscope (Zeiss Airyscan 2 LSM900) | Carl Zeiss AG | N/A |
| CFX96 thermocycler | Bio-rad | N/A |
| NextSeq 550 | Illumina | N/A |
| Chemoluminescence scanner (ECL Chemostar) | Intas Science Imaging Instruments GmbH | N/A |

## Reagents and antibodies

The following antibodies were used for immunofluorescence staining or Western blot analysis: monoclonal mouse/human anti-HDAg (Wang et al, 2021) (1:3000, FD3A7; commercially available), rabbit anti-HBcAg antibody (Ni et al, 2014) (1:1000, H363) and rabbit anti-L-HDAg (Lempp et al, 2019) (1:5000) were generated in-house. Human anti-HBsAg (1:1000, HBC34) was a kind gift from Davide Corti, Humabs BioMed. Rabbit anti-FoxA2 (1:400) was purchased from Cell Signaling, mouse anti-AFP (1:1000) from Sigma-Aldrich, mouse anti-ALB (1:1000) from Cedarlane, mouse anti-CD63 from Santa Cruz Biotechnology (1:400), rabbit anti-ADAR1 from Cell Signaling (1:1000), rabbit anti-LDLR from Abcam (1:200), rabbit anti-LAMP1 (1:200) and rabbit anti-EGFR (1:100) from Cell Signaling and rabbit anti-SCARB1 from Novus Biologicals (1:500). Alexa Fluor 488/

568 anti-mouse (1:1000), Alexa Fluor 488 anti-human (1:1000), and Alexa Fluor 488/568 anti-rabbit (1:1000) antibodies were purchased from Invitrogen. Antibodies for western blot: mouse anti-CD63 (1:1000) was purchased from Invitrogen, rabbit anti-PVR (1:1000) from Sigma-Aldrich, mouse anti-ITGβ1 (1:1000) from Santa Cruz and rabbit anti-ITGα2 (1:1000) from Abcam. Lonafarnib was purchased from Selleckchem.

## Standard cell culture

The human hepatoma cell line Huh7$^{NTCP}$ ectopically expressing human NTCP was generated previously (Ni et al, 2014). Huh7$^{NTCP}$ cells were cultured in Dulbecco's modified Eagle medium (DMEM, Gibco) supplemented with 10% advanced fetal bovine serum (FBS, Capricorn) and 2 µg/mL puromycin (InvivoGen). HepaRG cells were cultured and differentiated as previously described (Gripon et al, 2002). For AAV production, HEK-293 cells were cultured in DMEM (Gibco), supplemented with 10% fetal bovine serum and 1% penicillin–streptomycin (Gibco). All cell lines used in this study were routinely tested negative for mycoplasma.

## Generation of human pluripotent stem cell-derived hepatocyte-like cells (HLCs)

The hESC line WA09 (WiCell) was cultured in mTeSR1 medium (STEMCELL Technologies) on Matrigel (Corning) coated plates. WA09 cells were differentiated to definitive endoderm (DE) using the STEMdiff™ Definitive Endoderm Kit (STEMCELL Technologies) according to the manufacturer's protocol. To induce hepatic differentiation, DE cells were differentiated in the basal medium consisting of CTS™ KnockOut™ DMEM/F-12 (Gibco), 10% KnockOut Serum Replacement (KOSR, Gibco), 1% MEM solution of non-essential amino acids (NEAA, Gibco), 1% GlutaMAX supplement (Gibco), and 1% penicillin–streptomycin (Gibco), supplemented with human hepatocyte growth factor (HGF, Prepotech), Dimethylsulfoxide (DMSO, Sigma-Aldrich), and dexamethasone (Sigma) as previously described (Dao Thi et al, 2020). For final maturation, HLCs were cultured in the Hepatocyte Culture Medium BulletKit™ (HCM, Lonza) supplemented with 20 ng/mL of recombinant human oncostatin M (OSM, R&D systems). For the co-culture experiment, WA09 cells were transduced with lentivirus to express ZsGreen and selected with 1 µg/mL puromycin as described (Dao Thi et al, 2016) prior to HLC differentiation. prior to HLC differentiation.

## HBV and HDV production and infection

HBV (GT D) virus was produced from the HepF19 cell line as previously described (Wettengel et al, 2021). HDV virus was produced as previously described (Lempp et al, 2019). In brief, virus was collected from supernatants from Huh7 cells co-transfected with plasmids pJC126 (Gudima et al, 2002) (HDV GT1, kindly provided by John Taylor, Fox Chase Cancer Center) and pT7HB2.7 (Sureau et al, 1994) (hepatitis B virus genotype D envelope proteins, kindly provided by Camille Sureau, Institut National de la Transfusion Sanguine; accession number: MN645906) and purified by heparin affinity chromatography. HDV virus stocks of GT1 Ethiopia and 3–8 were produced as previously described (Wang et al, 2021).

HLCs were infected at day 18 of the differentiation protocol at an MOI of 5 infectious units (IU) in the presence of 4% polyethylene glycol 8000 (PEG, Sigma-Aldrich) and 1.5% DMSO (Sigma-Aldrich). In all, 16 to 24 h later, the inoculum was removed and HLCs were washed twice with Dulbecco's Phosphate Buffered Saline (DPBS) before replenishing with fresh culture medium supplemented with 1.5% DMSO. Medium was exchanged every two days until the end of the experiment.

For the secondary infection, $1 \times 10^5$ Huh7$^{NTCP}$ cells were seeded in 24-well plates and inoculated with approximately one-fifth of the culture supernatant from HDV-infected HLCs in DMEM containing 4% PEG and 2% DMSO (Carl Roth) after 24 h. 16 to 24 h post-infection, cells were washed twice with PBS and replenished with fresh DMEM supplemented with 2% DMSO. Medium was exchanged every two days until the end of the experiment.

## Adeno-associated virus (AAV) production and transduction

The hNTCP or CD63 gene was cloned into a self-complementary AAV vector (pscAAV-CMV-EYFP-BGHpolyA) (Börner et al, 2020). The HB2.7 subgenomic fragment encoding the L-/M-/S-HBsAg of HBV genotype D was cloned into a single-stranded AAV vector (pSSV9-AAV-CMV-EYFP-BGHpolyA) (Wolff et al, 2019). Recombinant AAVs of serotype 6 were produced and iodixanol-purified as described previously (Maurer et al, 2022). HLCs were transduced with AAV6s 2 days prior to HDV infection (to express NTCP or CD63) or one day after HDV infection (to express HBsAg). Twenty-four hours post transduction, the inoculum was removed, HLCs were washed with DPBS, and replenished with fresh culture medium.

## Biosafety

Experiments involving viral infections were conducted within a Biosafety Level 2 (BSL-2) facility at Heidelberg University and were approved by the "Regierungspräsidium Tübingen".

## Stem cell transfection

In all, 300 ng pJC126 (or mock) and 3 µl lipofectamine Stem Transfection Reagent (Invitrogen) were each mixed with 25 µl OptiMEM in two tubes. After incubating at room temperature (RT) for 5 min, the contents in both tubes were mixed and incubated for an additional 30 min at RT. WA09 cells were dissociated into single cells using Accutase, and $4 \times 10^4$ cells were seeded in mTeSR1 supplemented with 10 µM Y-27632 on Matrigel-coated wells of a 24-well plate. The transfection mix was added dropwise to the cells. Cells were passaged once and then fixed for immunofluorescence analysis nine days after transfection.

## Immunofluorescence (IF) staining and microscopy

For IF staining, cells were fixed with 4% paraformaldehyde (PFA, Science Services) for 15 min at RT. For the staining against HDAg, HBcAg, and HBsAg, cells were permeabilized with 0.5% TritonX-100 (Millipore) for 10 min, followed by the incubation with primary antibody diluted in 1% casein overnight at 4 °C. For differentiation marker staining (ALB and AFP) and CD63, cells were blocked and permeabilized in PBTG (10% goat serum, 1%

bovine serum, and 0.1% TritonX-100 in PBS) for 30 min prior to incubation with primary antibody diluted in PBTG overnight at 4 °C. After three washes with PBS, cells were incubated with secondary antibodies diluted either in 1% casein or PBTG for 1 h at RT. For NTCP staining, cells were fixed with 1.25% PFA, followed by overnight incubation with 1.5 µM Atto$^{488}$ or Atto$^{565}$ labeled Myrcludex B (Meier et al, 2013) (MyrB$^{Atto488/565}$) at 4 °C. Hoechst 33342 (1:1000, Thermo Fisher Scientific) was used for nuclei staining. Images were taken either on a Zeiss Cell discoverer 7 (CD7) microscope or a confocal microscope Zeiss Airyscan 2. For the quantification of HDV infection events, roughly 70% of cells grown in a well of a 24-well plate were imaged. Images were processed using ImageJ and ZEN software. HDV infection events were quantified using CellProfiler (Stirling et al, 2021).

## Quantitative reverse-transcription PCR (qRT-PCR)

Total RNA was isolated from cell lysates using the NucleoSpin RNA kit (Macherey-Nagel) and from cell supernatants using the QIAmp viral RNA mini kit (Qiagen) according to the manufacturers' protocols. The secondary structure of HDV RNA was denatured by heating RNA at 95 °C for 5 min followed by fast cooling to -80 °C. Reverse transcription was performed using the High-Capacity cDNA Reverse Transcription Kit (ABI). HDV RNA was detected using the Luna cbcUniversal Probe qPCR Master Mix (New England Biolabs) with the Ferns primers and probe listed in Appendix Table S1 on a CFX96 thermocycler (Bio-rad). The JC126 plasmid encoding HDV GT1, kindly provided by Dr. John Taylor, was used to generate a standard curve for quantifying HDV total RNA copy numbers. The expression of other genes was quantified using iTaq™ Universal SYBR Green Supermix (Bio-rad) with primers listed in Appendix Table S1. Both the HDV RNA copies and relative expression data were normalized to expression against the housekeeping gene RPS11.

## Enzyme-linked immunosorbent assay (ELISA)

Secreted hepatitis B surface antigen (HBsAg) was quantified in the cell supernatant by enzyme-linked immunosorbent assay (Architect, Abbott) according to the manufacturer's protocol. Absolute values above 0.05 International Units/mL were scored positive. Secreted HBV e antigen (HBeAg) levels were quantified by Advia Centaur XP Immunoassay System (Siemens). Values above 1 were considered positive.

## RNA-seq analysis

RNA isolated from biological duplicates using the NucleoSpin RNA kit (Macherey-Nagel) was used as input. RNA integrity was determined using Agilent RNA Nano 6000 chips on the Agilent Bioanalyzer 2100 system (Agilent Technologies). Sequencing libraries were constructed using the NEBNext Ultra II Directional RNA Preparation Kit (New England Biolabs) and the NEBNext Multiplex Oligos for Illumina. Samples were sequenced on the Illumina NextSeq 550 (150 cycles) in paired-end mode (paired-end sequencing) at the Deep Sequencing Core Facility of Heidelberg University.

RNA-seq reads were mapped to the human transcriptome assembly GRCh38 release 95 (also known as hg38) using Salmon version 0.15.2 (Patro et al, 2017) to estimate transcript abundance. Estimated abundances were then aggregated at the gene level using

txlmport version 1.24.0 (Soneson et al, 2015). Data normalization and differential expression analysis was performed using DESeq2 version 1.36.0 (Love et al, 2014). Finally, Gene Ontology (GO) enrichment analysis of the set of differentially expressed genes was performed using topGO version 2.48.0 (Alexa and Rahnenfuhrer, 2016).

## Custom-made siRNA screen

In all, 42 selected SMARTpool siRNAs (Dharmacon) were individually added to each well of a 96-well plate containing 20 µl OptiMEM and 0.2 µL Lipofectamine™ RNAiMAX Transfection Reagent at a final concentration of 12.5 or 50 nM for Huh7$^{NTCP}$ cells. Afterwards, $2 \times 10^4$ Huh7$^{NTCP}$ cells were added to each well in 100 µl culture medium. 48 h later, the supernatant was replaced with fresh medium, and cells were used for downstream experiments 24 h later.

## siRNA forward transfection after HDV infection

Overall, $1.5 \times 10^4$ Huh7$^{NTCP}$ cells were seeded per well of a 96-well plate in 120 µl culture medium. Cells were infected with HDV (MOI = 1) 24 h after cell seeding, followed by siRNA transfection on the next day. In all, 50 nM siRNA and 0.2 µl Lipofectamine™ RNAiMAX were each mixed with 10 µl OptiMEM in two tubes. After incubating at room temperature for 5 min, they were mixed and incubated for an additional 30 min at RT. Then, the transfection mix was added dropwise to the cells. After 48 h, the supernatant was removed and replaced with fresh medium. Twenty-four hours later, the cells were used for downstream experiments e.g., HDV infections.

## Western blot analysis

Cells were lysed in RIPA lysis and extraction buffer (Thermo Fisher Scientific) supplemented with 1× protease inhibitor cocktail (50× cOmpleteTM Mini Protease Inhibitor Cocktail, Roche). Cell lysates for NTCP detection were de-glycosylated by using PNGaseF (New England Biolabs) or untreated cell lysates were separated by 12% sodium dodecyl sulfate–polyacrylamide gel electrophoresis (SDS-PAGE). Proteins were electro-transferred onto a 0.45 µm poly-vinylidene fluoride (PVDF) or nitrocellulose membrane. The membrane was blocked in PBS containing 5% dry milk (Roth) for 1 h at RT and incubated with primary antibody overnight at 4 °C. After washing with PBS containing 0.1% Tween-20, the membrane was incubated with horseradish peroxidase-labeled goat-anti-mouse or goat-anti-rabbit antibodies (used at 1:4000, Jackson ImmunoResearch) for 1 h at RT. The membranes were imaged on the INTASELL Chemostar imager.

## Taurocholate uptake assay

HLCs were transduced with or without AAV encoding YFP or NTCP 2 days prior to the experiment. The taurocholate uptake assay was performed as described before (Lempp et al, 2019).

## Strand-specific detection of HDV genomes and antigenomes

HDV genomes (G) and antigenomes (AG) were detected using a modified method used by (Harichandran et al, 2019). For

quantitative analysis of the HDV G or AG, 500 ng of total cellular RNA was annealed with dNTP mix (10 mM) and the primer Tag1-G-RT-Taylor for the G or Tag2-AG-RT-Taylor for the AG at 95 °C for 5 min and then held at 60 °C. In all, 5× SSIV buffer, DTT (100 mM), RNaseOut RNase inhibitor and SuperScript IV reverse transcriptase (Thermo Fisher) were then added. The first-strand cDNA was generated at 60 °C for 10 min and reverse transcriptase was inactivated at 80 °C for 10 min. The cDNA product was incubated with RNase H (M0297S, New England Biolabs) at 37 °C for 20 min allowing a complete removal of the G/AG RNA hybrid to the first-strand DNA. The cDNA products were further purified using DNA Clean & Concentrate-5 (Zymo Research).

Strand-specific quantitative PCR was performed using iTaq™ Universal SYBR Green Supermix (Bio-rad) with the primers Tag1-G-F-Taylor and Ferns for the G, or Tag2-AG-F-Taylor and AG-R-1 for the AG as summarized in Appendix Table S1. Plasmid JC126 was used as the standard. The qPCR program was 95 °C for 3 min (denaturing) and 45 cycles of 95 °C for 10 sec and 60 °C for 40 sec (annealing and extension). Using in vitro transcribed G and AG, these assays exhibited a specificity of 25 for G-specific PCR and 208 for AG-specific PCR. The numbers provided by non-specific amplification of the other strand were subtracted in the raw copy numbers absolutely quantified and copy numbers were finally calculated per 1 µg total RNA.

## Detection of HBV cccDNA and total DNA

Total cellular DNA was extracted using Quick-DNA/RNA Micro-prep Plus Kit (Zymo Research). 500 ng of total DNA samples were subjected to T5 exonuclease digestion at 37 °C for 30 min. HBV cccDNA levels were determined using a cccDNA-specific qPCR as suggested by an evidence-based guidance (Allweiss et al, 2023) and primers and probe as described in Appendix Table S1. The qPCR program is 95 °C for 10 min, followed by 45 cycles of 95 °C for 15 s, 60 °C for 60 s and 72 °C for 10 s (Malmström et al, 2012). Undigested DNA samples were used for the detection of total HBV DNA and the human β-globin gene, which was used to normalize cccDNA copy numbers (Qu et al, 2018).

## Statistics

Graphs and statistical analyses were performed using GraphPad PRISM 8. In all figures where p-values were calculated, the corresponding statistical test is listed in the figure legend.

## Declaration of generative AI and AI-assisted technologies in the writing process

During the preparation of this work, the authors used DeepL Write in order to improve language and readability. After using this tool/service, the authors reviewed and edited the content as needed and take full responsibility for the content of the publication.

## Ethics approval statement

This project (AZ: 3.04.02/0179) has been approved by the Zentrale Ethik-Kommission für Stammzellenforschung of the Robert Koch Institute (RKI) and fulfils all legal requirements according to the German Stem Cell Act.

## Data availability

The RNA-seq data sets have been deposited in NCBI's Gene Expression Omnibus and are accessible through GEO Series accession number GSE270625.

The source data of this paper are collected in the following database record: biostudies:S-SCDT-10_1038-S44319-024-00236-0.

## Peer review information

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

## Acknowledgements

The authors gratefully acknowledge Dr. Volker Lohmann, Dr. John Taylor and Dr. Camille Sureau for generously sharing plasmids. We acknowledge Dr. Vibor Laketa, head of the Infectious Diseases Imaging Platform (IDIP) and Dr. David Ibberson, head of the Deep Sequencing Core Facility, at the University Heidelberg for their expert support. We thank Andrew Freistaedter, Franziska Schlund, Dr. Samara Martín Alonso and Dr. Vera Sonntag-Buck for excellent technical support. We thank Dr. Yi Ni and Dr. Benno Zehnder for valuable and helpful discussions. The graphical abstract was created using BioRender. For the publication fee we acknowledge financial support by Heidelberg University and SFB/TRR179. This work was supported by grants from the Deutsche Forschungsgemeinschaft (DFG, German Research Foundation)—Projektnummers 272983813—SFB/TRR 179 and 240245660—SFB1129; DZIF—TTU Hepatitis Projects 05.704., 05.806., and 05.823; Chica and Heinz Schaller Foundation. BQ was supported by BMBF grant VIRASCREEN 01KI2113. JH was supported by a fellowship from the China Scholarship Council. XW was supported by NIH grant DP2-AI170515.

## Author contributions

**Huanting Chi**: Conceptualization; Data curation; Software; Formal analysis; Validation; Investigation; Visualization; Methodology; Writing—original draft; Writing—review and editing. **Bingqian Qu**: Conceptualization; Data curation; Software; Formal analysis; Validation; Investigation; Visualization; Writing—review and editing. **Angga Prawira**: Data curation; Investigation; Methodology. **Talisa Richardt**: Formal analysis; Validation; Investigation; Methodology. **Lars Maurer**: Investigation; Methodology. **Jungen Hu**: Investigation; Methodology. **Rebecca M Fu**: Validation; Investigation. **Florian A Lempp**: Investigation. **Zhenfeng Zhang**: Resources; Investigation; Methodology. **Dirk Grimm**: Resources; Supervision; Methodology. **Xianfang Wu**: Conceptualization; Supervision; Methodology. **Stephan Urban**: Conceptualization; Supervision; Funding acquisition; Visualization; Project administration. **Viet Loan Dao Thi**: Conceptualization; Formal analysis; Supervision; Funding acquisition; Visualization; Writing—original draft; Project administration; Writing—review and editing.

Source data underlying figure panels in this paper may have individual authorship assigned. Where available, figure panel/source data authorship is listed in the following database record: biostudies:S-SCDT-10_1038-S44319-024-00236-0.

## Funding

## Disclosure and competing interests statement

Stephan Urban is a co-inventor and applicant on patents protecting HBV pre-S1-derived lipopeptides (Myrcludex B/Bulevirtide/Hepcludex). The remaining authors declare no competing interests.

# Expanded View Figures

**Figure EV1. HLCs are susceptible for HDV infection.** ▶

(A, B) HLCs were infected with HDV at different MOIs (0.5, 1, 2.5, or 5 Int. Units/cell) and HDV antigen levels were analyzed (HDAg, red) 5 days p.i. Scale bar = 100 μm. HDAg-positive cells were counted using CellProfiler. $N = 6$ biological replicates from two independent HLC differentiations. (C) HLCs, differentiated HepaRG, and Huh7$^{NTCP}$ cells were infected with HDV (MOI = 5 Int. Units/cell). HDAg-positive cells were counted using CellProfiler. HDV infection efficiency is shown either by infectious unit per mL (IU/mL) or HDV infection percentage $N = 5$ biological replicates from two independent experiments. (D) HLCs were transduced with or without AAV6-YFP or AAV6-NTCP. Two days post transduction, cell lysates were harvested and analyzed by Western blot analysis. (E) HLCs were infected with the indicated HDV genotype and 5 days p.i., cells were harvested for HDAg staining. Scale bar = 100 μm. (F) HLCs were infected with HDV (MOI = 5 Int. Units/cell) and harvested on indicated days p.i. HDV replication was analyzed by quantifying HDV genome copies in infected HLC lysates using RT-qPCR. Dashed line = LOQ. $N = 4$ biological replicates. (G) Huh7$^{NTCP}$ cells were infected with HDV (MOI = 5 Units/cell). RNA lysates were harvested on day 1, 3, 5, 7, 9 p.i. HDV copy numbers of genomes and antigenomes were determined by strand-specific qRT-PCR, respectively. $N = 7$ biological replicates from two independent experiments. (H) HLCs, Huh7$^{NTCP}$, and differentiated HepaRG cells were uninfected or infected with HDV (MOI = 1 5 Units/cell).) or treated with BLV (500 nM). Cellular protein lysates were collected on day 1, 3, 5, 10, 15 p.i. and levels of L- and S-HDAg were analyzed by Western blot. Data information: In (B, C, F and G) data are presented as mean ± SD. Statistical significance in (B) was tested among HDV-infected HLC at different MOIs ($P < 0.0001$) by ordinary one-way ANOVA. Statistical significance in (C) was tested between HDV-infected HLC and dHepaRG ($P_{IU/mL} = 0.1384$; $P_\% = 0.0875$) and between HDV-infected HLC and Huh7$^{NTCP}$ ($P_{IU/mL} < 0.0001$; $P_\% < 0.0001$) by multiple comparisons of ordinary one-way ANOVA. Statistical significance in (F) was tested between day 10- and day 15- harvested HDV-infected HLC ($P < 0.0001$) by multiple comparisons of ordinary one-way ANOVA. Statistical significance in (G) was tested between day 1- and day 3-harvested HDV-infected Huh7$^{NTCP}$ ($P_{gRNA} < 0.0001$; $P_{agRNA} < 0.0001$) by an unpaired two-tailed *t* test. ****$P < 0.0001$, n.s.: non-significant.

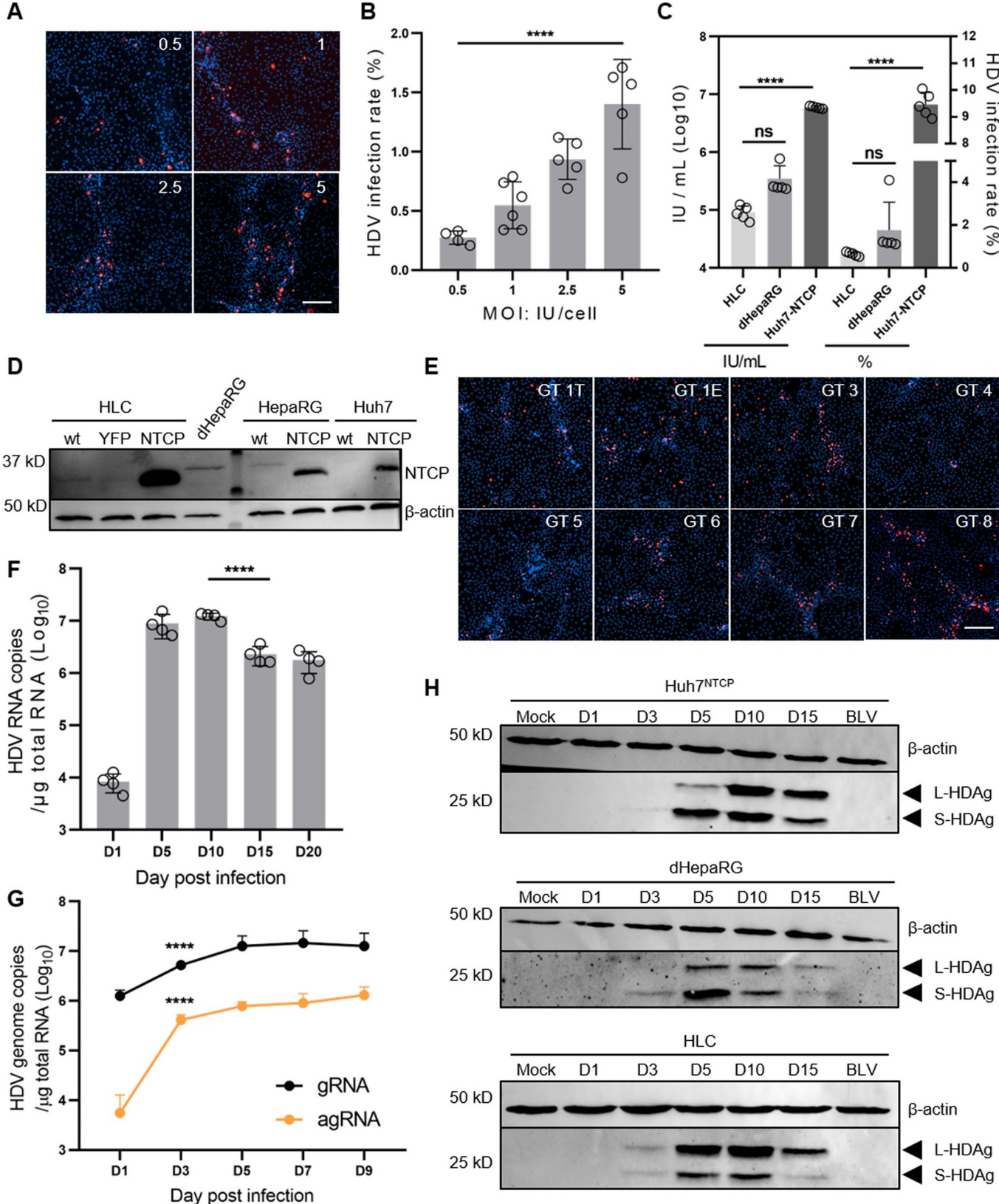

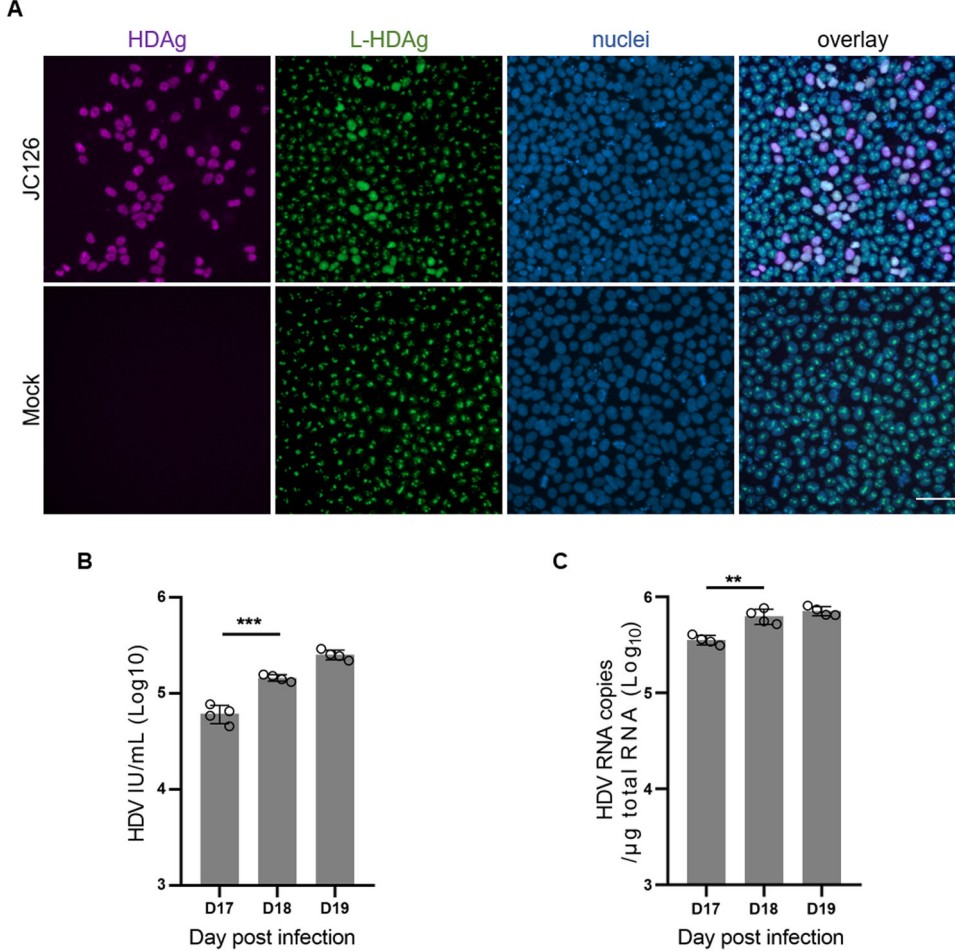

**Figure EV2. HDV susceptibility along D17 to D19 of the HLC differentiation protocol.**

(**A**) Undifferentiated WA09 cells were transfected with or without pJC126. Cells were harvested and assessed by immunofluorescence staining against the HDAg (magenta) and L-HDAg (green) 8 days post transfection. Scale bar = 50 μm. (**B**, **C**) HLCs were infected with HDV (MOI = 5) at the indicated day of the differentiation protocol and harvested for analyzing HDV infection efficiency by quantifying HDV-positive cells through CellProfiler (**B**) or detecting HDV total RNA copies through RT-qPCR (**C**) on 5 days p.i. $N = 4$ biological replicates from two independent HLC differentiations. Data information: In (**B**, **C**) data are presented as mean ± SD and statistical analysis was performed by multiple comparisons of ordinary one-way ANOVA. Statistical significance in (**B**) was tested between day 17- and day 18-harvested HDV-infected HLC ($P = 0.0004$). Statistical significance in (**C**) was tested between day 17- and day 18-harvested HDV-infected HLC ($P = 0.002$). ***$P < 0.001$; **$P < 0.01$.

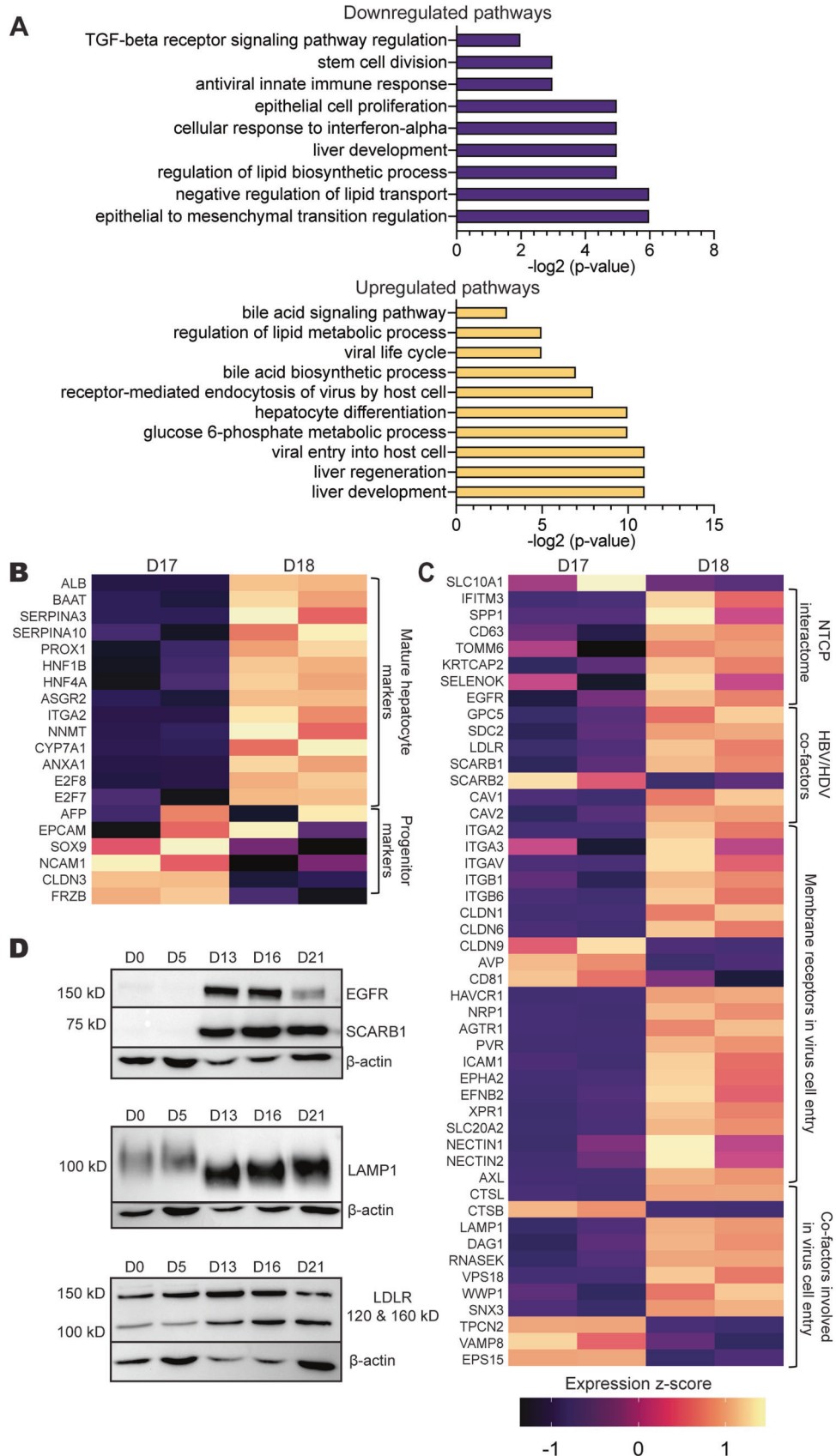

◀ **Figure EV3.  Gene expression along HLC differentiation.**

Total RNA was extracted from HLCs at either day 17 or 18 during the differentiation protocol and subjected to whole-transcriptome expression profiling and gene ontology (GO) term enrichment analysis. (A) GO term enrichment analysis of biological pathways for up- and downregulated genes between HLCs at day 17 and 18. Differentially expressed genes ($P$ value < 0.05) were significantly enriched in this GO term. Statistical analysis was performed using the Kolmogorov–Smirnov test. (B–D) Heatmap of Z score-normalized counts per million (CPM) values for (B) hepatocyte markers, and (C) virus entry factors. $N = 2$ biological replicates. (D) Cellular protein lysates of mature HLCs were collected and expression levels of previously described HBV/HDV entry factors and cofactors were analyzed by western blot.

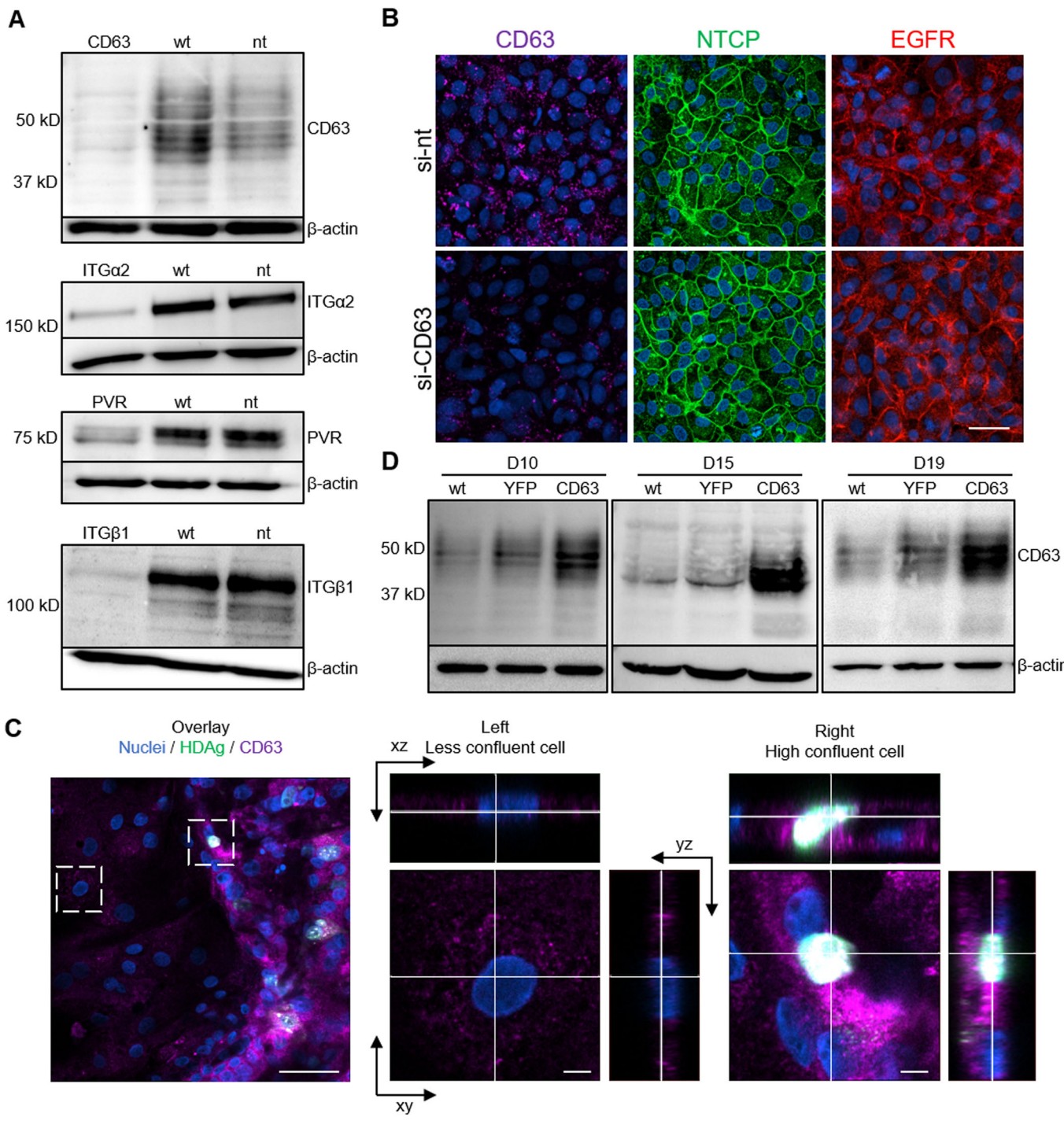

**Figure EV4. Knockdown of CD63 does not alter membranous NTCP expression.**

(A) Four siRNAs targeting CD63, ITGA2, PVR, and ITGB1 were delivered into Huh7^NTCP cells and downregulation was confirmed by Western blot analysis on cell lysates harvested 3 days post transfection. Untransfected (wt) and non-target (nt) siRNAs were used as controls. (B) Huh7^NTCP cells were transfected with 50 nM siRNA targeting CD63. 3 days later, NTCP protein stained with Atto-MyrB-488 (green), CD63 and EGFR stained with specific antibody were imaged by confocal microscopy. Scale bar = 50 μm. (C) Left: Mature HLCs were infected with HDV, fixed 5 days p.i., and stained for CD63 (magenta), HDAg (green), and the nucleus (blue). Images were taken on the Airyscan confocal microscope. Left: Shown is a 40x tile image with the two HLC populations (reused from Fig. 5G, merge). Scale bar = 50 μm. Middle & Right: Zoom in image of the region of interest. Single z-slice and orthogonal xz and yz views of less (middle) and high (right) confluent HLCs. Scale bar = 5 μm. (D) Endogenous and ectopic CD63 expression at the different hepatocyte differentiation stages. Cells were transduced with AAV6s two days before harvesting them at indicated day of the differentiation protocol. Lysates were analyzed by Western blot using specific anti-CD63 and β-actin antibodies.

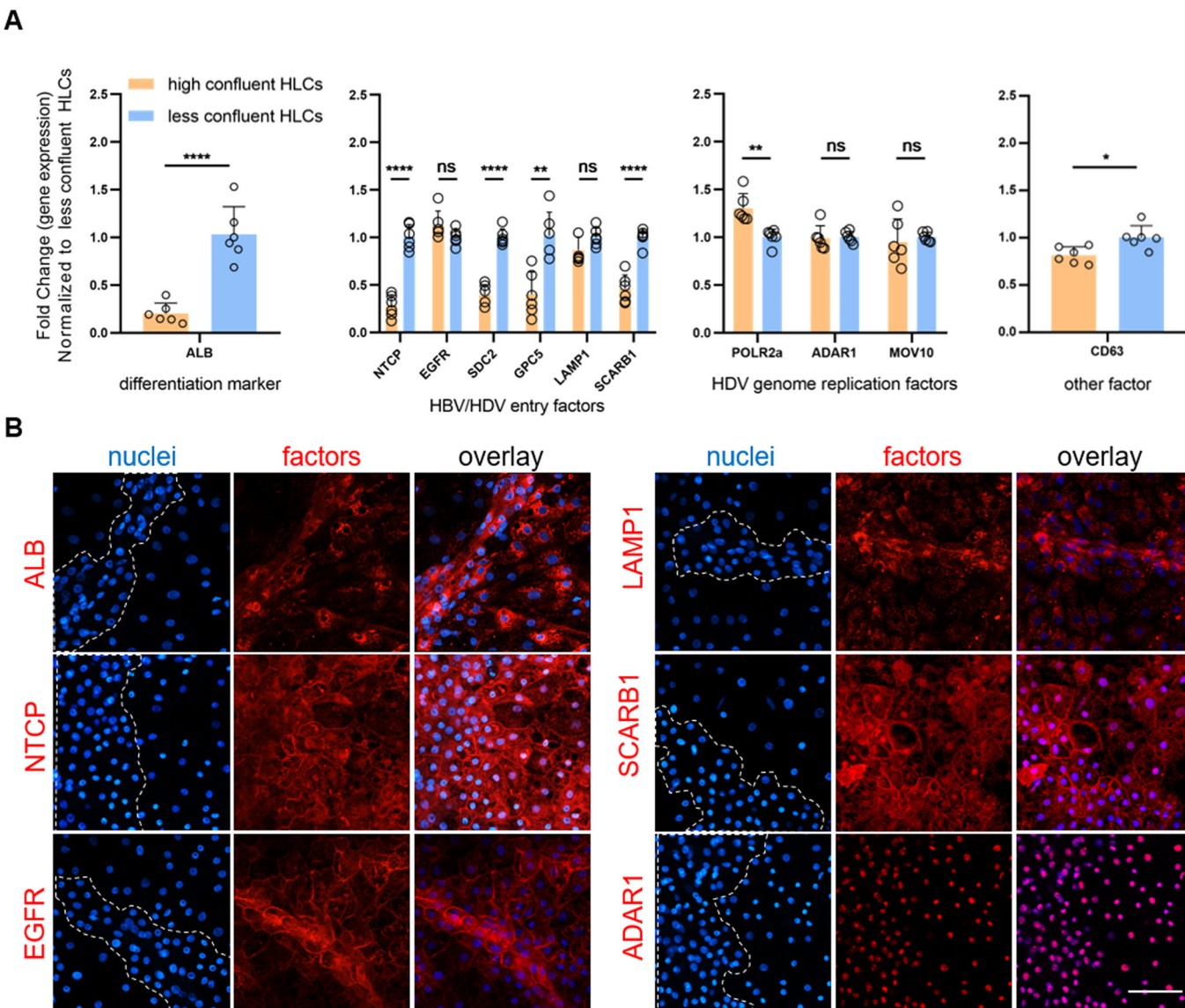

**Figure EV5. Transcriptional and protein levels of host factors in high and less confluent HLC populations.**

(A) Cellular lysates were harvested from high and less confluent HLC populations. Differentiation factor *ALB*, entry factors (*NTCP, EGFR, SDC2, GPC5, LAMP1, SCARB1*), HDV genome replication factor (*POLR2A, ADAR1, MOV10*) and other virus–host interaction factors (CD63) were analyzed using RT-qPCR. $N = 6$ biological replicates from two independent HLC differentiations. (B) Selected factors in (A) were immunostained in HLCs (shown in red). Images were taken on the Airyscan confocal microscope. Shown are 40× tile image with the two HLC populations. Scale bar = 50 μm. Highly confluent HLCs are indicated by the white dashed line. Data information: In (A) data are presented as mean ± SD and statistical analysis was performed by unpaired two-tailed $t$ test. Statistical significance was tested between high and less confluent HLC ($P_{ALB} < 0.0001$; $P_{NTCP} < 0.0001$; $P_{EGFR} = 0.0950$; $P_{SDC2} < 0.0001$; $P_{GPC5} = 0.0012$; $P_{LAMP1} = 0.0738$; $P_{SCARB1} < 0.0001$; $P_{POLR2a} = 0.0018$; $P_{ADAR1} = 0.08617$; $P_{MOV10} = 0.6186$; $P_{ALB} = 0.0117$). ****$P < 0.0001$; **$P < 0.01$; *$P < 0.05$; n.s.: non-significant.

