## [Peer Review File · EMBO Reports]

An Hepatitis B and D Virus Infection Model Using Human Pluripotent Stem Cell-Derived Hepatocytes

Viet Loan Dao Thi, Huanting Chi, Bingqian Qu, Angga Prawira, Talisa Richardt, Lars Maurer, Jungen Hu, Rebecca Fu, Florian Lempp, Zhenfeng Zhang, Dirk Grimm, Xianfang Wu, and Stephan Urban

Corresponding author(s): Viet Loan Dao Thi (vietloan.daothi@med.uni-heidelberg.de) , Stephan Urban (Stephan.Urban@med.uni-heidelberg.de)

Review Timeline:

Transfer from Review Commons:	23rd Jan 24
Editorial Decision:	25th Jan 24
Revision Received:	26th Jun 24
Editorial Decision:	25th Jul 24
Revision Received:	7th Aug 24
Accepted:	22nd Aug 24

Editor: Achim Breiling

**Transaction Report: This manuscript was transferred to
EMBO Reports following peer review at Review Commons.**

**Review
COMMONS**

Revision Plan

Manuscript number: RC-2023-02266

Corresponding author(s): Urban, Stephan; Dao Thi, Viet Loan

1. General Statements

We thank the Editors and the Reviewers for their time and constructive criticism, which has allowed us to improve our manuscript. All of our responses are indicated in blue font. Revision Figures for the Reviewers are included just below the response. The line numbers given here refer to those in the revised manuscript, where we have marked the changes in red.

2. Description of the planned revisions

If granted a full revision, we will experimentally address the following major points, which were raised by more than one Reviewer:

- Repeat experiment in Figure 4 C to assess statistical significance (Reviewer 1 and 3)
- Western blot analysis of HDV infected HLCs showing small and large delta antigens. We have already performed such an analysis on HLCs (see Revision Figure 2). In addition, we will perform a comparative analysis with common HDV infection models dHepaRG and Huh7-NTCP cells over time (Reviewers 2 and 3).
- Additional characterisation of the two HLC subpopulations at transcript and protein level (Reviewer 1 and 3).

In addition, we planned to conduct the following experiments in response to the individual Reviewers:

In response to Reviewer 1:

We thank the Reviewer for their encouraging feedback on our model and for their helpful comments, allowing us to improve our manuscript.

Figure 1: The observation of a denser subpopulation of hepatocytes more susceptible to HDV is interesting. Do you have more characterization of this cell subpopulation, by IFA, in term of hepatic maturation marker, known HDV host factors and particularly NTCP expression?

We agree with the Reviewer that this is an interesting observation. We separated the two hepatocyte subpopulations to analyse the gene expression of the liver maturation markers NTCP and ALB by RT-qPCR (see Revision Figure 1A). Surprisingly, we found that the low-density population expressed higher levels of both ALB and NTCP, suggesting that they are more mature than the high-density population. In addition, we stained both markers by immunofluorescence and observed no apparent differences (see Revision Figures 1B & C). In contrast, the new host factor identified in our study, CD63, appeared to be more highly expressed in the high-density population compared to the low-density population (Fig. 6G). However, we cannot exclude the

Revision Plan

possibility that other factors play an additional role. As outlined in our response to Reviewer 3, we will separate the two populations and analyse the gene expression of other known HBV and HDV co-host factors to assess whether they play a role in addition to CD63 in conferring the higher susceptibility to HDV infection to the highly dense HLC population.

Revision Figure 1: High-density HLCs population is not more mature than the low-density HLC population. (A) The low-density HLCs population was separated from the high-density HLC population by gentle dissociation. Total RNAs were isolated from both populations and Albumin and NTCP expression was analysed by RT-qPCR. (B & C) High-density HLCs (upper image) and low-density HLCs (bottom image) were stained with Albumin specific antibody. Shown are either images taken on an epifluorescence microscope (B) or single slices of confocal images acquired on a Airyscan confocal microscope (C).

Fig 1B and C: Can a BLV control be included in the figure?

Thank you for this suggestion, we will repeat the experiment for these panels and add BLV as control.

Fig 1A-F: What is the overall level of NTCP between HLC, HepaRG, Huh7NTCP and HLC-AAV-NTCP? Can NTCP and HDAg be stained simultaneously in your cells?

This is an excellent question and we will compare the total NTCP levels between differentiated HepaRG, Huh7 NTCP, HLCs +/- AAV NTCP by Western blot analysis and immunofluorescence (IF) staining. Comparing NTCP expression in HLCs +/- AAV NTCP, we observed a strong up-regulation of surface NTCP upon AAV transduction by IF staining (Figure 1D).

Unfortunately, our initial attempts to simultaneously detect NTCP and HDAg were technically hampered. Since HDAg is mainly localised in the nuclei, we have to permeabilize the cells in a harsh manner, which interferes with the detection of membrane NTCP. The latter is further hampered by the availability of suitable anti-NCTP antibodies for IF staining. In our study, we used high doses of fluorescence-conjugated MyrB peptide to stain NTCP, but unfortunately it is very sensitive to the harsh permeabilization detergents mentioned above. However, since we have meanwhile optimised HDV infection, we will likewise try again to optimise the staining

Revision Plan

protocol. If we succeed, we will repeat the co-staining of NTCP and HDAg and include it in a revised manuscript.

Figure 4: While the strategy is interesting, based on what has been previously shown for HCV in Wu et al., 2012, the lack of statistical data prevents the reader to really understand and see drastic difference in term of susceptibility to infection and level of expression of host genes. In panel C, is the difference between day 13 and 15 statistically significant? Same for panel D, day 17 vs 19? As a remark, day 19, the peak of susceptibility to HDV, seems to be also the peak of maturation, based on ALB RTqPCR (panel B).

Thank you for this comment, and will perform another set of experiments allowing us to calculate statistical significance. The Reviewer correctly points out the correlation between HDV infection and hepatocyte maturity, which we find very intriguing. To identify potential host co- or restriction factors expressed in highly mature HLCs, we then performed the differential gene expression analysis (Figure 5). As shown in the new Figure 5A, GO analysis revealed that genes involved in pathways regulating viral entry into host cells were most significantly upregulated in mature HLCs and, as a probable consequence, they were more permissive to HDV infection. Indeed, among these factors, we identified CD63 as a novel host cofactor that renders mature HLCs susceptible to HDV infection (Figure 6).

In response to Reviewer 2:

We thank the Reviewer for their assessment of our study and for critically pointing out the increments over the previous study by Lange et al. We also appreciate their helpful suggestions, which allow us to improve the manuscript.

The manuscript would benefit from a more detailed virological analysis, such as:

•Determination of HDV genome and antigenome sequences and analysis of HDV editing.

We thank the Reviewer for this comment. Accordingly, we will determine HDV genomes and antigenomes by Northern blot analysis and study HDV editing rates by sequencing in HDV-infected HLCs.

•Analysis of HDV short and large antigens by western blot.

We have already detected small and large HDAg in HDV-infected HLCs (see Revision Figure 2). To also satisfy Reviewer 3, we will additionally compare the S/L-HDAg ratios over time in HLCs, dHepaRGs, and Huh7-NTCP cells and include the results in a revised manuscript.

Revision Figure 2: Detection of small and large delta antigen in HDV-infected HLCs. Mature HLCs were infected with HDV (MOI= 5 Int. Units/cell) and harvested 1 or 3 days post-infection. Cell lysates were analysed by Western blotting using antibodies against HDAg and b-actin.

Revision Plan

•Analysis of HBV-related virological parameters in monoinfected and co-infected cells.

We agree with the Reviewer and we will include the characterisation of more HBV-related virological parameters in our mono- and co-infected HLCs. Accordingly, we will assess HBV cccDNA, RNA, and DNA by RT-qPCR, as well as released HBsAg and HBeAg via ELISA and add the results to the revised manuscript.

In response to Reviewer 3:

We thank the Reviewer for their positive evaluation, and we acknowledge their helpful comments, which will help us to improve our manuscript.

Line 143: the authors describe two forms of HLCs (less and more confluent with differences regarding the susceptibility to HDV infection). The characteristics of the less and more confluent HLCs should be described in more detail-what is causative for the differences in susceptibility for HDV infection of these two forms?

We thank the Reviewer for this comment. We likewise find this observation intriguing. As stated in our response to Reviewer 1, we have ruled out that NTCP and/or other mature markers such as ALB are differentially expressed between the two subpopulations. As one factor that could make a difference, we have identified CD63, which is highly expressed in the high-density HLC population and less so in the low-density HLC population (Figure 6G). Nevertheless, we will separate the two populations and analyse by RT-qPCR the expression of other known HBV and HDV host co-factors that may be additional factors governing the increased susceptibility of the highly dense HLC population.

The statistical analyses should be improved: There are no p-values provided for the data presented in the supplement and a variety of figures lacks p-values

We have added p-values to the Supplementary Figures (see revised Supplementary Fig. S2) and will repeat the experiments for Fig. 4 and Supplementary Fig. S1B and Fig S3 so that we can calculate the corresponding p-values.

Kinetic of the infection: Here it would be interesting to see a comparative analysis by western blot investigating the ratio HBsAg/HDAg over the time in HLCs, HepaRGs and NTCP oe cells

We thank the Reviewer for his comments. As stated in our response to Reviewer 2, we will perform this WB analysis to detect S/L-HDAg over time in infected HLCs, dHepaRG, and Huh7-NTCP cells.

Line 157: What is the experimental evidence for the proper localization and functionality of the ectopically expressed NTCP in HLCs. Did the authors study the taurocholate transport after overexpression of NTCP?

We thank the Reviewer for this comment. We analysed endogenous and ectopic NTCP expression by microscopy using a fluorescently conjugated peptide Atto-MyrB-565, which specifically binds to the ectodomain of human NTCP (Figure 2D) and found that both

Revision Plan

endogenously and ectopically expressed NTCP are located on the cell surface. To further confirm the correct localisation, we will perform NTCP co-staining with a cell membrane marker.

We will also test the proper function of the ectopically expressed NTCP using a specific taurocholate transport assay as shown in our previous study (Ni et al, 2014, Gastroenterology).

Line 169: The authors should include data comparing the number of double positive cells in HLCs, HepaRGs and Huh7NTCP o.e. expressing cells under the chosen experimental conditions

We thank the Reviewer for this suggestion. We have already performed HBV/HDV co-infection of dHepaRG cells (Revision Figure 3) and we will perform the same experiment with Huh7-NTCP cells.

Revision Figure 3: HBV/HDV co-infection of dHepaRG cells. Differentiated HepaRG were infected with HBV (MOI = 450 genome copies/cell) and HDV (MOI = 5). Cells were stained against HBV core (HBc), HDAg, and nuclei (DAPI) ten days p.i.. HBc- and HDAg-positive cells were counted using Cellprofiler imaging software to quantify HBV (pink) and HDV (green) single and co-infection (white) events. Images are representative of three independent differentiations.

Line 291: expression analysis by RT-PCR is not sufficient. It will be important to study by CLSM if the identified factors are really present as proteins and properly localized.

To satisfy this Reviewer, we will be happy to perform WB analysis of lysates from cells obtained at different stages of HLC differentiation to detect LDLR, LAMP1 and SR-B1 to further confirm our transcriptome analysis. As protein expression is easier to compare by WB analysis, we prefer this method to microscopic analysis.

Regarding the role of CD63: what is the evidence for a direct role of CD63 for HDV entry- can the authors exclude that CD63 is relevant for targeting other factors to the surface? What is the impact of loss of CD63 on the functionality of the autophagosomal-MVB-EV system in HLCs?

Since downregulation of CD63 before but not after impairs HDV infection, we conclude that CD63 is likely to be important for the early steps of the HDV life cycle, namely cell entry. Indeed, we speculate that CD63 may be critical for HDV trafficking to the vesicle, where fusion of the HBV glycoproteins is induced to allow capsid entry, based on the following observations: Although neither the precise site of HBV viral fusion nor the cues that induce fusion are currently fully understood, studies suggest that HBV can be co-transported with EGFR and NTCP to late endosomes for trafficking (Herrscher et al; 2020, Cells). We speculate that similar to what has been described for Lujo virus, CD63 may be involved in either HDV trafficking and/or virus fusion in the endosomal system (late endosome or lysosome) (Tominaga et al, 2014 Molecular cancer).

CD63 is a ubiquitously expressed protein that localises to the endosomal system and, in its glycosylated form, to the cell surface. Non-glycosylated CD63 is not properly trafficked and aggregates at the nuclear periphery instead of the cell membrane (Tominaga et al., 2014, Molecular Cancer). According to the Western blot analysis in Figure 6, immature HLCs appear to express less glycosylated CD63 than mature HLCs. We will confirm the glycosylation by treating the cell lysates with PNGase F. Although AAV transduction enhanced CD63 expression of all three HLC stages tested (see new Supplementary Figure S6 in the revised manuscript), it only enhanced HDV infection of immature HLCs, in which the non-glycosylated form of CD63 appears to be the predominant form. To demonstrate that the glycosylated form of CD63 is involved in HDV entry, we will rescue WT CD63 in parallel with a glycosylation-deficient CD63 mutant (Yoshida et al., 2009, Microbiology and Immunology) in immature HLCs. We will also stain CD63 in both immature and mature HLCs to compare the subcellular localisation (plasma membrane/endosomes vs. nuclear membrane) of CD63 between the two stages.

3. Description of the revisions that have already been incorporated in the transferred manuscript

Based on the constructive comments by the Reviewers we already made the following changes, which are highlighted in red in the revised manuscript.

In response to Reviewer 1:

Fig 1B-C: the comparison with dHepaRG is very interesting, and confirms the validity of SC derived hepatocytes as a model for HDV infection. dHepaRG can be heterogeneous. Do you also see the same phenotype of enriched HDV infection within a denser subpopulations of dHepaRG

We thank the Reviewer for their comment. Undifferentiated bipotent HepaRG cells are not permissive for HDV infection due to the lack of surface NTCP expression. Due to their bipotent nature and upon differentiation, two morphologically distinct populations become apparent: hepatocyte-like cells and biliary epithelial-like cells (McGill et al., 2010, Hepatology). As shown in the Figure 1 of the study by Mesnage et al. (2018, Molecular Toxicology), dense hepatocyte-like colonies are surrounded by clear epithelial cells corresponding to primitive biliary cells. In agreement with other studies, we only observe that the ALB-positive hepatocyte-like cells are permissive to HBV and HDV infection (Hantz et al., 2009, Journal of General Virology), highlighting their specific hepatic tropism and the cellular determinants required.

Fig 1I is confusing. Was BLV assay also performed on the HLC infection (Day 0), or only during the titration assay in Huh7NTCP?

We apologise for the confusion in this panel. BLV was only added during the titration assay on Huh7NTCP cells to confirm new and productive infections and to rule out carry-over. We have changed the order of Figures 1I - 1K to make this clearer and explain this better in the new results section (line 171-179) and figure legend (line 797-806).

Revision Plan

Fig 1K: x-axis is confusing... is it number of HBV, HDV and HBV/HDV positive cells? Or number of infected cells upon inoculation with HBV, HDV, or both? Please clarify.

We apologise for this additional confusion caused in this panel. We infected HLCs with both HBV and HDV simultaneously and then counted the number of positive cells that were either single infected with HBV (pink cells/column), single infected with HDV (green cells/column) or double infected with both viruses (white cells/column). We have clarified this in the revised Results section (line 172-176) and in the revised Figure Legend (line 798-803).

Figure 2: The AAV based vector to over express HBsAg is a very interesting tool, and the figure convincingly show production of HDV progeny viruses in HLC-AAV-HBsAg. Results shown are in agreement with previous studies based on hepatoma cell lines.

We thank the Reviewer for this positive comment and we agree that AAVs represent interesting tools to genetically manipulate HLCs and other hepatocyte culture systems.

Figure 2B: What is IU/ml? Infectious Unit? International Unit? Are units in Fig 1B, 2B and 2C the same?

We apologise for the lack of clarity. In Figures 1B and 2C, IU corresponds to infectious units of HDV, whereas in Figure 2B, IU corresponds to international units for the assessment of secreted HBsAg levels in the supernatant. To make the difference clearer, we have changed the unit on the y-axis in Figure 2B and explicitly stated the abbreviations in the corresponding revised Figure Legends (lines 785, 786, 794, 795, 816, and 819).

Figure 3: What is the overall number of transmission events observed in the co-culture setup? Can you visually observed viral spreading? Panel A shows only 1 event, making it hard to assess its efficiency. Titration assay in Fig 2C show production of up to 4-5 log of infectious HDV. But HLCs susceptibility to HDV infection may change during time...

Thank you for your comment and for raising this important issue. Panel A clearly and visually demonstrates that extracellular spread of HDV had occurred in the HLCs system, as initially only WT and non-GFP positive HLCs were infected with HDV. After co-culture, the progeny of WT HLCs were able to infect GFP-HLCs (Figure 3A). The overall efficiency of HDV spread/transmission in HLC efficiency is shown in Figure 3C. If we allow spread to occur (DMSO-treated condition), the total number of HDV-positive HLCs grown in a 24-well plate is approximately 1000. When we block secondary infection of progeny with BLV and thus spread, we count only about 500 HDV-positive HLCs in a well.

In general, spreading in HLCs (Figure 3C) is not as efficient as re-infection to Huh7-NTCP (Figure 2C) for the following reasons: In Figure 2C, we wanted to have an estimation of the maximum amount of secreted infectious progeny from HDV-producing HLCs. To this end, we did not want the re-infection itself to be a major bottleneck and used the most susceptible model Huh7-NTCP and infected them under the best conditions, which includes the addition of 4% PEG and 2% DMSO in the culture medium.

For our spread assay in HLCs, we cannot add PEG to the cells over the course of the experiment and we also wanted to be as physiological as possible. PEG significantly enhances HDV infection

Revision Plan

of HLCs (Supplementary Fig. S2) and Huh7-NTCP cells (Revision Figure 4), which is in agreement with previous studies (Michailidis et al., 2017, Scientific reports). In addition, as the Reviewer correctly points out, similar to other primary hepatocyte culture models, the HLC system deteriorates over time. However, we have found that HLCs can be cultured for up to 3 weeks. Nevertheless, we believe that the efficiency of HDV spread in HLCs is sufficient for drug testing (Fig. 3C & D).

Revision Figure 4: PEG enhances HDV infection of Huh7-NTCP cells. Huh7-NTCP cells were infected with HDV (MOI= 5 Int. Units/cell) in the absence or presence of PEG. Cells were harvested on D5 pi and HDV genome copies were quantified by RT-qPCR.

Figure 5: In panel A, GO pathways should be sorted based on significance, not Number of genes. In panel B-D, what is the scale of the heatmap on figure 5: change in CPM values, however log2, log10?

Thank you for this comment, we have sorted the GO pathways based on significance (new Figure 5A). For panels B-D, we did not calculate the fold change in CPM values and they were not log transformed. Instead, we calculated the z-scores of the genes shown by comparing the expression level of a given gene (in CPM) in a given sample with the expression level of that gene across all samples. To avoid further confusion, we have added "z-score" to the new Figure 5.

Figure 6: Do you have info about CD63 in other mature model, like dHepaRG and PHHs? Is CD63 also limiting in these models?

Our data in Figure 6 suggest that CD63 may be a limiting factor for HDV infection of immature HLCs but not mature HLCs. Both dHepaRG cells and PHHs are mature hepatocyte models and therefore we speculate that CD63 is not rate limiting. However, we will investigate whether CD63 is rate-limiting in undifferentiated HepaRG cells.

In response to Reviewer 2:

Additional information that needs to be added, better explained, or corrected:

The authors should explain why they used different MOIs depending on the genotype.

In our previous study by Wang et al. 2021 J Hepatol, we found that the different HDV genotypes are heterogeneous in their ability to infect Huh7 NCTP cells. For example, as shown in Figure 4B of Wang et al. 2021 J Hepatol, GT 4 and 5 are less infectious than other genotypes. Based on the different infectious titres of the genotypes obtained on Huh7 NTCP cells, we then decided to use different MOIs for infection of our HLCs. The aim of the present study by Chi et al. was not to

Revision Plan

compare the different HDV genotypes, but to analyse whether they can all infect HLCs. In order to obtain similar infection efficiencies of our HLCs with the different genotypes, we used higher MOIs for those genotypes that were less infectious in Huh7-NTCP cells compared to those genotypes that were more infectious in Huh7-NCTP cells. We apologise for not making this sufficiently clear and have added this information to the results section (line 167-170) and the corresponding figure legend (line 796) of the revised manuscript.

In Figure 1, it is unclear on which day the HCLs were infected by HDV and on which day they were transduced with AAV-NTCP.

We apologise for the lack of clarity in the experimental design. We transduced HLCs with AAV two days before HDV infection to ensure sufficient ectopic NTCP expression on the day of HDV infection to study its effect on HDV entry. We have clarified this in the results section (line 153, 156) and in the figure legend (line 788) in the revised manuscript.

It is not very clear if the authors used AAV serotype 6 consistently to transduce the cells. It would be valuable to show the transduction efficiency of AAV at different time points of HLC maturation, as it might also be affected and could explain some results. For example, in Figure 6H, why does AAV-CD63 transduction increase HDV infectivity at day15 but not at day 10? It would be interesting to repeat the anti-CD63 western blot after AAV-CD63 transduction.

Thank you for this comment. Yes, we have consistently used AAV 6 due to its relatively broad tissue tropism (Verdera et al., 2020, Molecular Therapy) and we have clarified this information in the revised manuscript (see line 331). We agree with the Reviewer's concerns regarding the variable transduction efficiency. We have previously tested different AAV capsids and found that AAV6 transduced mature HLCs at high levels (Zhang et al., 2022, Hepatol Commun). In this study, we also performed Western blot analysis to confirm successful CD63 overexpression by AAV transduction at different stages of hepatocyte differentiation. As shown in new Supplementare Figure 6, although there were some differences in transduction efficiency, the majority of all cells at each stage of differentiation were successfully transduced to ectopically express CD63.

The authors claim that by using AAV to express HBsAg, they are mimicking the expression of HBsAg from the integrated sequence rather than cccDNA. However, it is the opposite, as AAV genomes, like cccDNA, remain as episomes in the cells.

Yes, the Reviewer is conceptually correct and we apologise for the incorrect wording. In principle, we aim to trans-complement HBsAg in a setting outside of HBV infection and thus mimic the expression of antigen from integrated cells, although AAVs of course remain mostly episomal. We have clarified this in the revised manuscript (see lines 188 & 378).

In response to Reviewer 3:

Line 217: the complete inhibition of cell to cell spread by myrcludex suggests that there is no spread by cell-cell contact. This should be discussed.

Revision Plan

Yes, there is no evidence of HDV spread by cell-cell contact because, as the Reviewer correctly points out, BLV treatment almost completely blocked HDV de novo infection (Figure 2D & E). To our knowledge, cell-to-cell spread has not been demonstrated for HDV. According to our own studies by Zhang et al, 2021/2022, Journal of Hepatology, HDV spreads either extracellularly (which can be blocked by BLV) or by cell division (discussed in lines 362). Since HLC are similar to primary human hepatocytes and do not divide in vitro, we believe that extracellular spread is the predominant mode of spread in HLC (line 365).

Line 210ff: Is there any evidence for syncytia formation in this system?

No, we have not observed syncytia formation. Since HDV has no glycoproteins, we would not expect syncytia to form.

Line 42: secrete should be replaced by release

We thank the Reviewer for pointing out the inaccuracy in our terminology. We have replaced "secrete" with "release" (line 42).

Line 241: proteins are not expressed, genes are expressed

Thank you, we agree and changed the wording accordingly (line 246).

4. Description of analyses that authors prefer not to carry out

In response to Reviewer 1:

Fig 1B: Unit is confusing, using terms usually used for titration of infectivity, from the virus input point of view, not from the cellular point of view. Can you use % infected cells instead, or "HDV infection rate" like in Supp Fig 1B?

We apologise for this confusion. For other viruses, such as but not limited to HCV or HEV, the most common method is to report focus forming units per ml (FFU/ml). HLCs do not divide and, in the absence of HBV S antigen, no cell-division mediated HDV spread can occur and only single infection events can be observed (hence infectious unit = IU/ml). Since differentiated, authentic hepatocyte culture models such as PHHs, HLCs or HepaRG cells are always characterised by strong cell heterogeneity, it is difficult to directly compare the overall percentage of infection with a homogeneous cell population such as Huh7-NTCP cells. Therefore, if the Reviewer allows us, we prefer to keep this unit in our main figures. However, and hopefully to the satisfaction of this Reviewer, we have also calculated the percentage of infected cells of this exact dataset and show it in the Supplementary figures (Suppl. Fig. S1 C). The proportion of infection efficiency comparing HLCs, dHepaRGs, and Huh7-NTCP cells does not differ when presented either as IU/ml or as percentage of infected cells.

Dear Dr. Dao Thi,

Thank you for the transfer of your research manuscript from Review Commons to EMBO reports. I now went through your manuscript, the referee reports from Review Commons (attached again below) and your revision plan. The referees have several comments, concerns, and suggestions to improve the manuscript, indicating that a major revision of the manuscript is necessary to allow publication of the study.

Going through your revision plan, it seems that most of these points will be adequately addressed during revision. I thus invite you to revise your manuscript accordingly with the understanding that all concerns must be addressed in the revised manuscript and/or in a detailed point-by-point response (as indicated in your revision plan). Acceptance of your manuscript will depend on a positive outcome of another round of review using the same set of referees.

It is EMBO reports policy to allow a single round of major revision only and acceptance of the manuscript will therefore depend on the completeness of your responses included in the next, final version of the manuscript.

- 1) a .docx formatted version of the final manuscript text (including legends for main figures, EV figures and tables), but without the figures included. Figure legends should be compiled at the end of the manuscript text.
- 2) individual production quality figure files as .eps, .tif, .jpg (one file per figure), of main figures (up to 8) and EV figures (up to 5). Please upload these as separate, individual files upon re-submission.

For more details, please refer to our guide to authors:
<http://www.embopress.org/page/journal/14693178/authorguide#manuscriptpreparation>

Please consult our guide for figure preparation:
http://wol-prod-cdn.literatumonline.com/pb-assets/embo-site/EMBOPress_Figure_Guidelines_061115-1561436025777.pdf

See also the guidelines for figure legend preparation:
<https://www.embopress.org/page/journal/14693178/authorguide#figureformat>

3) a final .docx formatted letter INCLUDING the reviewers' reports and your detailed point-by-point responses to their comments. As part of the EMBO Press transparent editorial process, the point-by-point response is part of the Review Process File (RPF), which will be published alongside your paper.

4) a complete author checklist, which you can download from our author guidelines (<https://www.embopress.org/page/journal/14693178/authorguide>). Please insert page numbers in the checklist to indicate where the requested information can be found in the manuscript. The completed author checklist will also be part of the RPF.

Please also follow our guidelines for the use of living organisms, and the respective reporting guidelines:
<http://www.embopress.org/page/journal/14693178/authorguide#livingorganisms>

5) that primary datasets produced in this study (e.g. RNA-seq, ChIP-seq, structural and array data) are deposited in an

appropriate public database. If no primary datasets have been deposited, please also state this in a dedicated section (e.g. 'No primary datasets have been generated and deposited'), see below.

The accession numbers and database should be listed in a formal "Data Availability" section (placed after Materials & Methods) that follows the model below. This is now mandatory (like the COI statement). Please note that the Data Availability Section is restricted to new primary data that are part of this study. This section is mandatory. As indicated above, if no primary datasets have been deposited, please state this in this section

Data availability

8) Regarding data quantification and statistics, please make sure that the number "n" for how many independent experiments were performed, their nature (biological versus technical replicates), the bars and error bars (e.g. SEM, SD) and the test used to calculate p-values is indicated in the respective figure legends (also for potential EV figures and all those in the final Appendix). Please also check that all the p-values are explained in the legend, and that these fit to those shown in the figure. Please provide statistical testing where applicable. Please avoid the phrase 'independent experiment', but clearly state if these were biological or technical replicates. Please also indicate (e.g. with n.s.) if testing was performed, but the differences are not significant. In case n=2, please show the data as separate datapoints without error bars and statistics. See also: <http://www.embopress.org/page/journal/14693178/authorguide#statisticalanalysis>

9) Please also note our reference format:

10) We updated our journal's competing interests policy in January 2022 and request authors to consider both actual and perceived competing interests. Please review the policy <https://www.embopress.org/competing-interests> and update your competing interests if necessary. Please name this section 'Disclosure and Competing Interests Statement' and put it after the Acknowledgements section.

11) We now use CRediT to specify the contributions of each author in the journal submission system. CRediT replaces the author contribution section. Please use the free text box to provide more detailed descriptions and do not provide an author contributions section in the revised manuscript text file. See also guide to authors:

<https://www.embopress.org/page/journal/14693178/authorguide#authorshippinguidelines>

12) Please add scale bars of similar style and thickness to all the microscopic images, using clearly visible black or white bars (depending on the background). Please place these in the lower right corner of the images themselves. Please do not write on or near the bars in the image but define the size in the respective figure legend.

13) Please provide a final title with not more than 100 characters (including spaces).

- 14) Please provide the abstract written in present tense throughout.
- 15) Please reduce the number of keywords to five.
- 16) Please remove the list of abbreviations from the manuscript text file. Make sure that each abbreviation is defined the first time used.
- 17) Please add the funding information to the Acknowledgement section. Please make sure that all the funding information is also entered into the online submission system and that it is complete and similar to the one in the acknowledgement section of the manuscript text file.
- 18) Please add all the material and methods information to the main manuscript text file.
- 19) We would encourage you to use 'Structured Methods', our new Materials and Methods format. According to this format, the Materials and Methods section should include a Reagents and Tools Table (listing key reagents, experimental models, software and relevant equipment and including their sources and relevant identifiers), uploaded as separate file, followed by a Methods and Protocols section in which we encourage the authors to describe their methods using a step-by-step protocol format with bullet points, to facilitate the adoption of the methodologies across labs. More information on how to adhere to this format as well as downloadable templates (.doc or .xls) for the Reagents and Tools Table can be found in our author guidelines (section 'Structured Methods'):

- 20) Please order the manuscript sections like this, using these names:
Title page - Abstract - Keywords - Introduction - Results - Discussion - Materials and Methods - Data availability section - Acknowledgements - Disclosure and Competing Interests Statement - References - Figure legends - Expanded View Figure legends

Please note that all corresponding authors are required to supply an ORCID ID for their name upon submission of a revised manuscript. Please find instructions on how to link the ORCID ID to the account in our manuscript tracking system in our Author guidelines: <http://www.embopress.org/page/journal/14693178/authorguide#authorshipguidelines>

I look forward to seeing a revised version of your manuscript when it is ready. Please let me know if you have questions or comments regarding the revision.

Best,

Achim Breiling
Senior editor
EMBO reports

Referee #1:

In the present manuscript, H Chi et al describe the infection of stem cell derived hepatocytes with HBV and HDV. They suggest that it could be used to validate antiviral treatment in a mature hepatocyte model. Moreover, they take advantage of the differentiation process of the cells to identify time points correlating with significant change in viral permissivity, and focus on one of these time points in an attempt to identify new host factors of HDV.

Overall, the manuscript is well written and brings interesting information toward the establishment of an efficient HBV HDV coinfection model in stem cell derived hepatocytes. Particularly, comparison to dHepaRG, another model relying on in vitro differentiation and commonly used to study HBV and HDV infection, reveals the potential of stem cell derived hepatocytes. While the efficiency of co infection in the stem cell derived hepatocytes may seem low, the manuscript goes in the direction of helping establishing a new mature in vitro model of infection.

Figure 1: The observation of a denser subpopulation of hepatocytes more susceptible to HDV is interesting. Do you have more characterization of this cell subpopulation, by IFA, in term of hepatic maturation marker, known HDV host factors and particularly NTCP expression?

Fig 1B-C: the comparison with dHepaRG is very interesting, and confirms the validity of SC derived hepatocytes as a model for HDV infection. dHepaRG can be heterogeneous. Do you also see the same phenotype of enriched HDV infection within a

denser subpopulations of dHepaRG?

Fig 1B: Unit is confusing, using terms usually used for titration of infectivity, from the virus input point of view, not from the cellular point of view. Can you use % infected cells instead, or "HDV infection rate" like in Supp Fig 1B?

Fig 1B and C: Can a BLV control be included in the figure?

Fig 1A-F: What is the overall level of NTCP between HLC, HepaRG, Huh7NTCP and HLC-AAV-NTCP? Can NTCP and HDV be stained simultaneously in your cells?

Fig 1I is confusing. Was BLV assay also performed on the HLC infection (Day 0), or only during the titration assay in Huh7NTCP?

Fig 1K: x-axis is confusing... is it number of HBV, HDV and HBV/HDV positive cells? Or number of infected cells upon inoculation with HBV, HDV, or both? Please clarify.

Figure 2: The AAV based vector to over express HBsAg is a very interesting tool, and the figure convincingly show production of HDV progeny viruses in HLC-AAV-HBsAg. Results shown are in agreement with previous studies based on hepatoma cell lines.

Figure 2B: What is IU/ml? Infectious Unit? International Unit? Are units in Fig 1B, 2B and 2C the same?

Figure 3: What is the overall number of transmission events observed in the co-culture setup? Can you visually observed viral spreading? Panel A shows only 1 event, making it hard to assess its efficiency. Titration assay in Fig 2C show production of up to 4-5 log of infectious HDV. But HLCs susceptibility to HDV infection may change during time...

Figure 4: While the strategy is interesting, based on what has been previously shown for HCV in Wu et al., 2012, the lack of statistical data prevents the reader to really understand and see drastic difference in term of susceptibility to infection and level of expression of host genes. In panel C, is the difference between day 13 and 15 statistically significant? Same for panel D, day 17 vs 19?

As a remark, day 19, the peak of susceptibility to HDV, seems to be also the peak of maturation, based on ALB RTqPCR (panel B).

Figure 5: In panel A, GO pathways should be sorted based on significance, not Number of genes. In panel B-D, what is the scale of the heatmap on figure 5: change in CPM values, however log2, log10?

Figure 6: Do you have info about CD63 in other mature model, like dHepaRG and PHHs? Is CD63 also limiting in these models?

****Significance:****

Overall, the manuscript brings interesting information toward the establishment of an efficient HBV HDV coinfection model in stem cell derived hepatocytes. Particularly, comparison to dHepaRG, another model relying on in vitro differentiation and commonly used to study HBV and HDV infection, reveals the potential of stem cell derived hepatocytes. While the efficiency of co infection in the stem cell derived hepatocytes may seem low, the manuscript goes in the direction of helping establishing a critical needed and long awaited mature in vitro model of HDV HBV infection.

Referee #2:

In the present work, Chi et al. demonstrated that Hepatocyte-like cells (HCLs) derived from human pluripotent cells (hPCs) can be infected by HBV. The development of new HDV cellular models is of great value for understanding HDV biology and developing new treatments. However, the relevance of the present work is limited by a recent publication by Lange et al., in which they also showed that HCLs derived from hPCs can be infected by HDV, inducing the activation of the innate immune response, as previously demonstrated in cells and mice.

The authors added new information to the work of Lange et al, including:

- HLCs derived from human pluripotent cells can be infected by different HDV genotypes.
- They proved that infectious HDV particles are formed.
- They identified CD63 as a potential HDV coreceptor.

The manuscript would benefit from a more detailed virological analysis, such as:

- Determination of HDV genome and antigenome sequences and analysis of HDV editing.
- Analysis of HDV short and large antigens by western blot.

- Analysis of HBV-related virological parameters in monoinfected and co-infected cells.

Additional information that needs to be added, better explained, or corrected:

The authors should explain why they used different MOIs depending on the genotype.

In Figure 1, it is unclear on which day the HCLs were infected by HDV and on which day they were transduced with AAV-NTCP.

It is not very clear if the authors used AAV serotype 6 consistently to transduce the cells. It would be valuable to show the transduction efficiency of AAV at different time points of HLC maturation, as it might also be affected and could explain some results.

For example, in Figure 6H, why does AAV-CD63 transduction increase HDV infectivity at day 15 but not at day 10? It would be interesting to repeat the anti-CD63 western blot after AAV-CD63 transduction.

The authors claim that by using AAV to express HBsAg, they are mimicking the expression of HBsAg from the integrated sequence rather than cccDNA. However, it is the opposite, as AAV genomes, like cccDNA, remain as episomes in the cells.

****Significance:****

In the present work, Chi et al. demonstrated that Hepatocyte-like cells (HCLs) derived from human pluripotent cells (hPCs) can be infected by HBV. The development of new HDV cellular models is of great value for understanding HDV biology and developing new treatments. However, the relevance of the present work is limited by a recent publication by Lange et al., in which they also showed that HCLs derived from hPCs can be infected by HDV.

The authors added new information to the work of Lange et al, including:

- HLCs derived from human pluripotent cells can be infected by different HDV genotypes.
- They proved that infectious HDV particles are formed.
- They identified CD63 as a potential HDV coreceptor.

Referee #3:

In their manuscript entitled " An HBV/HDV Infection Model Using Human Pluripotent Stem 1 Cell-Derived Hepatocyte- Like Cells for Virus Host Interactions and Antiviral Evaluation" describe the use of HLCs derived from hPSCs as infection model for analysis of HDV life cycle. The ms is well written and clearly structured. It is easy to follow the concept of the study. The ms addresses a relevant topic and could help to overcome limitations in the analysis of HDV life cycle. The authors perform in many points a detailed characterization of this experimental system but there are still a variety of open points which must be addressed:

****Specific points:****

Line 143: the authors describe two forms of HLCs (less and more confluent with differences regarding the susceptibility to HDV infection). The characteristics of the less and more confluent HLCs should be described in more detail-what is causative for the differences in susceptibility for HDV infection of these two forms?

The statistical analyses should be improved: There are no p-values provided for the data presented in the supplement and a variety of figures lacks p-values

Kinetic of the infection: Here it would be interesting to see a comparative analysis by western blot investigating the ratio HBsAg/HDAg over the time in HLCs, HepaRGs and NTCP oe cells

Line 157: What is the experimental evidence for the proper localization and functionality of the ectopically expressed NTCP in HLCs. Did the authors study the taurocholate transport after overexpression of NTCP?

Line 169: The authors should include data comparing the number of double positive cells in HLCs, HepaRGs and NTCP o.e. expressing cells under the chosen experimental conditions

Line 217: the complete inhibition of cell to cell spread by myrcludex suggests that there is no spread by cell-cell contact. This should be discussed.

Line 210ff: Is there any evidence for syncytia formation in this system?

Line 291: expression analysis by RT-PCR is not sufficient. It will be important to study by CLSM if the identified factors are really present as proteins and properly localized.

Regarding the role of CD63: what is the evidence for a direct role of CD63 for HDV entry-can the authors exclude that CD63 is

relevant for targeting other factors to the surface? What is the impact of loss of CD63 on the functionality of the autophagosomal-MVB-EV system in HLCs?

****Minor points:****

Line 42: secrete should be replaced by release

Line 241: proteins are not expressed, genes are expressed

****Significance:****

The manuscript describes the use of HLCs derived from hPSCs as infection model for analysis of HDV life cycle. The ms is well written and clearly structured. It is easy to follow the concept of the study.

The ms addresses a relevant topic and could help to overcome limitations in the analysis of HDV life cycle. The authors perform in many points a detailed characterization of this experimental system but there are still a variety of open points which must be addressed.

Point-by-point reply to the reviewers' comments

Reviewer 1

Summary: In the present manuscript, H Chi et al describe the infection of stem cell derived hepatocytes with HBV and HDV. They suggest that it could be used to validate antiviral treatment in a mature hepatocyte model. Moreover, they take advantage of the differentiation process of the cells to identify time points correlating with significant change in viral permissivity, and focus on one of these time points in an attempt to identify new host factors of HDV.

Overall, the manuscript is well written and brings interesting information toward the establishment of an efficient HBV HDV coinfection model in stem cell derived hepatocytes. Particularly, comparison to dHepaRG, another model relying on in vitro differentiation and commonly used to study HBV and HDV infection, reveals the potential of stem cell derived hepatocytes. While the efficiency of co infection in the stem cell derived hepatocytes may seem low, the manuscript goes in the direction of helping establishing a new mature in vitro model of infection.

We thank the reviewer for their encouraging feedback on our model and for their helpful and constructive comments, allowing us to improve our manuscript.

Figure 1: The observation of a denser subpopulation of hepatocytes more susceptible to HDV is interesting. Do you have more characterization of this cell subpopulation, by IFA, in term of hepatic maturation marker, known HDV host factors and particularly NTCP expression?

We agree with the reviewer that this is an interesting observation. Accordingly, we separated the two hepatocyte subpopulations to analyse the expression of hepatocyte maturation markers NTCP and ALB, as well as known HDV host factors (according to a recent study by Lange et al., 2023, Liver Int.) by RT-qPCR (new Fig EV5A). Surprisingly, we found that the low-density population expressed higher levels of both ALB and NTCP, suggesting that they might be more mature than the high-density population. HDV host factors were generally equally expressed in both populations on the transcript level. To complement this analysis, we also analysed the protein expression of the different factors by immunofluorescence staining (new Fig EV5B). This analysis revealed that indeed CD63 (new Fig EV4C) together with other entry factors such as LAMP1, SCARB1 and EGFR appeared to be higher expressed in the highly dense HLC population. We have added these new findings to the revised result section (lines 366-370).

Fig 1B-C: the comparison with dHepaRG is very interesting, and confirms the validity of SC derived hepatocytes as a model for HDV infection. dHepaRG can be heterogeneous. Do you also see the same phenotype of enriched HDV infection within a denser subpopulations of dHepaRG

Yes, differentiation of bipotent HepaRG cells usually results in two morphologically distinct populations: hepatocyte-like cells and biliary epithelial-like cells (McGill et al., 2010, Hepatology). As shown in Figure 1 of the study by Mesnage et al. (2018, Molecular Toxicology), dense hepatocyte-like colonies are usually surrounded by clear epithelial cells corresponding to primitive biliary cells. We observe HDV infection only in these dense hepatocyte-like colonies (Revision Figure 1). In our HLCs, however, we found that both populations are hepatocyte-like cells, as evidenced by the expression of albumin and other hepatocyte markers (new Fig EV5A). Thus, the underlying causes of the difference in permissiveness to HDV between low and high density populations are likely to be different from those in dHepaRG cells.

Revision Figure 1: HDV infection of dense HepaRG derived hepatocyte-like colonies

HDV infection (MOI = 5) of dHepaRG cells with or without 500 nM entry inhibitor bulevirtide (BLV) was assessed by immunofluorescence staining (IF) against the HDV antigen (HDAg, red) five days post-infection (p.i.). Scale bar = 100 μ m.

Fig 1B: Unit is confusing, using terms usually used for titration of infectivity, from the virus input point of view, not from the cellular point of view. Can you use % infected cells instead, or "HDV infection rate" like in Supp Fig 1B?

We apologise for this confusion. For other viruses, such as, but not limited to, hepatitis C or E viruses, the most common practice is to report focus forming units per ml (FFU/ml). HLCs do not divide and, in the absence of the HBV surface antigen, no HDV spread can occur and only single infection events can be observed (hence, infectious unit = IU/ml). Since differentiated, authentic hepatocyte culture models such as PHHs, HLCs or HepaRG cells are always characterised by strong cell heterogeneity, it is difficult to directly compare the overall percentage of infection with a homogeneous cell population such as Huh7^{NTCP} cells, which additionally support cell-division mediated HDV spread (Zhang et al., 2022, J Hepatol). Therefore, if the reviewer allows us, we prefer to keep this unit in our main figures. However, we have also calculated the percentage of infected cells of this exact dataset as suggested by the reviewer. The results are shown in the extended figures (new Fig EV1C). The proportion of infection efficiency comparing HLCs, dHepaRGs, and Huh7^{NTCP} cells does not differ when presented either as IU/ml or as percentage of infected cells.

Fig 1B and C: Can a BLV control be included in the figure?

Thank you for this suggestion, we have repeated the experiments for these panels and added BLV as control (new Fig 1B, C).

Fig 1A-F: What is the overall level of NTCP between HLC, HepaRG, Huh7NTCP and HLC-AAV-NTCP? Can NTCP and HDAg be stained simultaneously in your cells?

This is an excellent question and we have compared NTCP levels between HLCs +/- NTCP, dHepaRG, HepaRG +/- NTCP, and Huh-7 +/- NTCP by Western blot analysis (new Fig EV1D). Importantly, we could only detect endogenous NTCP in HLCs, but not in dHepaRG, or Huh-7 WT cells. The ectopic NTCP expression was the strongest in HLCs but also clearly visible in HepaRG^{NTCP} and Huh-7^{NTCP} cells. Finally, yes, we were able to detect NTCP and HDAg simultaneously in HDV-infected HLCs (Revision Figure 2).

Revision figure 2: Simultaneous staining of NTCP and HDVAg in HDV-infected HLCs.

HLCs were infected with HDV (MOI=5) and stained against HDVAg (cyan) and NTCP (magenta) five days p.i. Scale bar = 10 μ m

Fig 1I is confusing. Was BLV assay also performed on the HLC infection (Day 0), or only during the titration assay in Huh7NTCP?

We apologise for the confusion in this panel. BLV was only added during the titration assay on Huh7^{NTCP} cells to confirm new infections and to rule out carry-over of the initial inoculum. We have changed the order of previous Figures 1I (now Fig 2G) to make this clearer and explain this better in the new results section (lines 225 and 227) and corresponding figure legends.

Fig 1K: x-axis is confusing... is it number of HBV, HDV and HBV/HDV positive cells? Or number of infected cells upon inoculation with HBV, HDV, or both? Please clarify.

We apologise for the confusion caused by this panel (now Fig. 2B). We infected HLCs with both HBV and HDV simultaneously and then counted the number of cells that were either single infected with HBV (pink cells/column), single infected with HDV (green cells/column) or double infected with both viruses (white cells/column). We have better clarified this in the results section (lines 21-213) and in the figure legend (lines 1076-1077).

Figure 2: The AAV based vector to over express HBsAg is a very interesting tool, and the figure convincingly show production of HDV progeny viruses in HLC-AAV-HBsAg. Results shown are in agreement with previous studies based on hepatoma cell lines.

We thank the reviewer for this positive comment and we agree that AAVs represent interesting tools to genetically manipulate HLCs and other hepatocyte culture systems.

Figure 2B: What is IU/ml? Infectious Unit? International Unit? Are units in Fig 1B, 2B and 2C the same?

We apologise for the lack of clarity. In Fig 1B and previous Fig 2C, IU corresponded to infectious units of HDV, whereas in previous Fig 2B, IU corresponded to international units for the assessment of secreted HBsAg levels in the supernatant. To make the difference clearer, we have changed the unit on the y-axis in the previous Fig 2B (now Fig 3B) and explicitly stated the abbreviations in the corresponding figure legends.

Figure 3: What is the overall number of transmission events observed in the co-culture setup? Can you visually observed viral spreading? Panel A shows only 1 event, making it hard to assess its efficiency. Titration assay in Fig 2C show production of up to 4-5 log of infectious HDV. But HLCs susceptibility to HDV infection may change during time...

Thank you for your comment and for raising this important issue. Panel A visually demonstrates that extracellular spread of HDV had occurred in the HLCs system, as initially only WT-HLCs were infected with HDV. After co-culture, the progeny from WT-HLCs were able to infect ZsGreen-HLCs (now Fig 3F).

The overall efficiency of extracellular HDV spread is moderate and can be assessed in previous Fig 3C (now Fig 3H). If we allow spread to occur (DMSO-treated condition), the total number of HDV-positive HLCs grown in a 24-well plate is approximately 1000

(normalised to 1, now Fig 3H). When we blocked secondary infection of progeny with BLV and thus spread, we counted only about 500 HDV-positive HLCs in a 24-well, meaning that roughly half of the final infection events detected were the result of HDV extracellular spread.

In general, extracellular HDV spread in HLCs is not as efficient as re-infection on Huh7^{NTCP} cells (now Fig 3C) for the following possible reasons: In Fig 3C, we wanted to have an estimation of the maximum amount of secreted infectious progeny from HDV-producing HLCs. To this end, we did not want the re-infection itself to be a major bottleneck and used Huh7^{NTCP} cells, which are the most susceptible cells (new Fig 1B, C). This titration had been performed under the best conditions, which includes the addition of 4% PEG and 2% DMSO in the culture medium. PEG significantly enhances HDV infection of HLCs (Revision Fig 3A) and Huh7^{NTCP} cells (Revision Fig 3B), which is in agreement with previous studies (Michailidis et al., 2017, Scientific reports). For our HDV spread assay in HLCs, we did not add PEG to the cells over the entire course of the experiment in order to avoid any negative impact on the HLCs.

In addition, as the reviewer correctly points out and similar to other primary hepatocyte culture models, the HLC system deteriorates over time. However, we have found that HLCs can be cultured for up to 3 weeks. We have included this limitation to the new Discussion section of the revised manuscript (lines 495-497).

Revision Figure 3: HDV infection of HLCs and Huh7^{NTCP} cells in the presence and absence of PEG.

A, B (A) HLCs and (B) Huh7^{NTCP} cells were infected with HDV (MOI=5) in the presence or absence of 4% PEG 8000. Cells were harvested 5 days p.i. and analyzed for HDV infections by counting HDAg-positive cells using CellProfiler (left panels) or quantifying HDV total RNA copies by RT-qPCR (right panels). N = 3 biological replicates. Statistical analysis was performed by unpaired two-tailed Student's t test ***: $p < 0.001$; ****: $p < 0.0001$;

Figure 4: While the strategy is interesting, based on what has been previously shown for HCV in Wu et al., 2012, the lack of statistical data prevents the reader to really understand and see drastic difference in term of susceptibility to infection and level of expression of host genes. In panel C, is the difference between day 13 and 15 statistically significant? Same for panel D, day 17 vs 19? As a remark, day 19, the peak of susceptibility to HDV, seems to be also the peak of maturation, based on ALB RTqPCR (panel B).

We thank the reviewer for encouraging us to repeat this experiment. Accordingly, we performed another set of experiments and found that the difference in HDV susceptibility between days 17 and 19 is highly significant whereas the difference between days 13 and 15 is not (now Fig 4C, D).

The reviewer correctly points out the correlation between HDV infection and hepatocyte maturity, which we found very intriguing. To identify potential HDV host co- or restriction factors expressed in highly mature HLCs, we then performed the differential gene expression analysis (now Fig EV3). The GO analysis revealed that genes involved in pathways regulating viral entry into host cells were most significantly upregulated in mature HLCs and, as a probable consequence, they were more permissive to HDV infection. Indeed, among these factors, we identified CD63 as a novel host cofactor that renders mature HLCs susceptible to HDV infection (now Fig 5).

Figure 5: In panel A, GO pathways should be sorted based on significance, not Number of genes. In panel B-D, what is the scale of the heatmap on figure 5: change in CPM values, however log2, log10?

Thank you for this helpful comment. We have sorted the GO pathways based on significance (now Fig EV3A). For panels B-D, we did not calculate the fold change in CPM values and they were not log transformed. Instead, we calculated the z-scores of the genes shown by comparing the expression level of a gene (in CPM) in a given sample with the expression level of that gene across all samples. To avoid further confusion, we have added "z-score" to the now Fig EV3.

Figure 6: Do you have info about CD63 in other mature model, like dHepaRG and PHHs? Is CD63 also limiting in these models?

This is indeed an interesting question. Since our data suggest that CD63 is a limiting factor for HDV infection of immature but not mature HLCs, we speculate that it is not rate-limiting in neither dHepaRG nor PHH, which are both representing mature hepatocyte models. We did however try to analyse if CD63 is a rate-limiting factor in undifferentiated HepaRGs ectopically expressing NTCP. As shown in Revision Figure 4, CD63 does not appear to be rate-limiting in this cell type.

Revision Figure 4: CD63 is not rate-limiting in undifferentiated HepaRG cells

- A Undifferentiated HepaRG or HepaRGNTCP cells were transduced to ectopically express CD63 and YFP respectively. CD63 overexpression was confirmed by Western blot analysis. wt: wildtype
- B Undifferentiated wt, YFP or CD63 overexpressed HepaRG or HepaRGNTCP cells were infected with HDV (MOI = 1). HDV infection was analyzed by counting HDAG-positive cells using CellProfiler. IU= infection unit

Reviewer #2

In the present work, Chi et al. demonstrated that Hepatocyte-like cells (HCLs) derived from human pluripotent cells (hPCs) can be infected by HBV. The development of new HDV cellular models is of great value for understanding HDV biology and developing new treatments. However, the relevance of the present work is limited by a recent publication by Lange et al., in which they also showed that HCLs derived from hPCs can be infected by HDV, inducing the activation of the innate immune response, as previously demonstrated in cells and mice.

The authors added new information to the work of Lange et al, including:

- HLCs derived from human pluripotent cells can be infected by different HDV genotypes.
- They proved that infectious HDV particles are formed.
- They identified CD63 as a potential HDV coreceptor.

We thank the reviewer for their assessment of our study and for pointing out the increments over the previous study by Lange et al. We also appreciate their helpful suggestions, which allowed us to improve the manuscript.

The manuscript would benefit from a more detailed virological analysis, such as:

- Determination of HDV genome and antigenome sequences and analysis of HDV editing.
- Analysis of HDV short and large antigens by western blot.
- Analysis of HBV-related virological parameters in monoinfected and co-infected cells.

We thank the reviewer for encouraging us to provide a more detailed virological analysis of both HBV and HDV-infected cells. Accordingly, we have quantified HDV genomes and antigenomes over the course of HDV-infection in HLCs (new Fig 1J). In addition, we have performed WB analysis to compare S/L-HDAg ratios over time in infected HLCs (new Fig 1K), dHepaRGs, and Huh-7^{NTCP} cells (new Fig EV1H). Finally, we also have performed additional experiments to quantify HBVcccDNA and RNA, as well as secreted HBsAg and HBeAg from HBV and HDV mono-infected as well as from HBV and HDV co-infected cells (new Fig 2C-F). In agreement with a previous study by Tham et al., 2020. Cell Rep Med, and since only 8% of the HBV-infected cells were co-infected, we did not observe any major changes in HBV parameters upon co-infection with HDV.

Additional information that needs to be added, better explained, or corrected:

The authors should explain why they used different MOIs depending on the genotype.

In our previous study by Wang et al., 2021 J Hepatol, we found that the different HDV genotypes are heterogeneous in their ability to infect Huh7^{NTCP} cells. For example, as shown in Figure 4B of Wang et al., 2021 J Hepatol, GT 4 and 5 are less infectious than other genotypes. Based on the different infectious titres of the genotypes obtained on Huh7^{NTCP} cells, we then decided to use different MOIs for infection of our HLCs. The aim of the present study by Chi et al. was only to assess if these genotypes can all infect HLCs, but not the underlying differences in their infectivity. In order to obtain similar infection efficiencies of our HLCs, we then used higher MOIs for those genotypes that were less infectious in Huh7^{NTCP} cells compared to those genotypes that were more infectious in Huh7^{NTCP} cells. We apologise for not making this sufficiently clear and have added this information to the results section (lines 189-191) and the corresponding figure legend of the revised manuscript.

In Figure 1, it is unclear on which day the HCLs were infected by HDV and on which day they were transduced with AAV-NTCP.

We apologise for the lack of clarity. We transduced HLCs with AAV two days before HDV infection to ensure sufficient ectopic NTCP expression on the day of HDV infection to study its effect on HDV entry. We have clarified this in the results section (lines 175-176) and in the corresponding figure legend of the revised manuscript.

It is not very clear if the authors used AAV serotype 6 consistently to transduce the cells. It would be valuable to show the transduction efficiency of AAV at different time points of HLC maturation, as it might also be affected and could explain some results. For example, in Figure 6H, why does AAV-CD63 transduction increase HDV infectivity at day15 but not at day 10? It would be interesting to repeat the anti-CD63 western blot after AAV-CD63 transduction.

Thank you for this comment. Yes, we used AAV6 throughout due to its relatively wide tissue tropism (Verdera et al., 2020, Molecular Therapy) and we clarified this information in the revised manuscript throughout by updating the result section and figure legends. We agree with the concerns of the reviewer about the potentially varying transduction efficiency. We have previously tested different AAV capsids and found that AAV6 transduces mature HLCs to high levels (Zhang et al., 2022, Hepatol Commun). We have now also performed a Western blot analysis to confirm successful and similar levels of CD63 overexpression by AAV transduction at the different hepatocyte differentiation stages (new Fig EV4D).

The authors claim that by using AAV to express HBsAg, they are mimicking the expression of HBsAg from the integrated sequence rather than cccDNA. However, it is the opposite, as AAV genomes, like cccDNA, remain as episomes in the cells.

Yes, the reviewer is conceptually correct and we apologise for the incorrect wording. In principle, we aim to trans-complement HBsAg in a setting outside of HBV infection and thus mimic the expression of antigen from integrated cells, although AAVs certainly remain mostly episomal. We have clarified this in the revised manuscript (lines 429-431).

Reviewer #3

In their manuscript entitled " An HBV/HDV Infection Model Using Human Pluripotent Stem 1 Cell-Derived Hepatocyte- Like Cells for Virus Host Interactions and Antiviral Evaluation" describe the use of HLCs derived from hPSCs as infection model for analysis of HDV life cycle. The ms is well written and clearly structured. It is easy to follow the concept of the study. The ms addresses a relevant topic and could help to overcome limitations in the analysis of HDV life cycle. The authors perform in many points a detailed characterization of this experimental system but there are still a variety of open points which must be addressed.

We thank the reviewer for their positive assessment and acknowledge their helpful comments, which helped us to improve our manuscript.

Line 143: the authors describe two forms of HLCs (less and more confluent with differences regarding the susceptibility to HDV infection). The characteristics of the less and more confluent HLCs should be described in more detail-what is causative for the differences in susceptibility for HDV infection of these two forms?

We thank the reviewer for this comment and also find this observation intriguing. As stated in our response to Reviewer 1, we have ruled out that NTCP and/or other mature markers such as ALB are differentially expressed between the two subpopulations (new Fig EV5). In addition, we found that the highly dense HLC population expressed higher levels of previously reported HBV/HDV cell entry factors such as LAMP1, SCARB1 and EGFR (new Fig EV5B), aside from CD63 which we identified in this manuscript (now Fig 5E). Therefore, we propose that CD63 together with these factors may render this HLC subpopulation more permissive to HDV infection. We have added these new findings to the revised result section (lines 367-370).

The statistical analyses should be improved: There are no p-values provided for the data presented in the supplement and a variety of figures lacks p-values

Thank you for sharing this concern. We have repeated the experiments for Fig 4C & D and supplemental Fig S1B (now Fig EV1B), Fig S3B (now Fig EV2B), Fig S3C (now Fig EV2C) allowing us to calculate the corresponding p-values.

Kinetic of the infection: Here it would be interesting to see a comparative analysis by western blot investigating the ratio HBsAg/HDAg over the time in HLCs, HepaRGs and NTCP oe cells

We thank the reviewer for this comment. As stated in our response to Reviewer 2, we have performed WB analysis to detect S/L-HDAg over time in infected HLCs, HepaRG, and Huh-7^{NTCP} cells (new Fig 1K and new Fig EV1H).

Line 157: What is the experimental evidence for the proper localization and functionality of the ectopically expressed NTCP in HLCs. Did the authors study the taurocholate transport after overexpression of NTCP?

We analysed endogenous and ectopic NTCP expression by microscopy using a fluorescently conjugated peptide Atto-MyrB-565, which specifically binds to the ectodomain of human NTCP (Fig 1D), and found that both endogenously and ectopically expressed NTCP were located on the cell surface. In addition, we have confirmed the proper functionality of the ectopically expressed NTCP by a specific taurocholate transport assay (Ni et al., 2014, Gastroenterology) (new Fig 1E).

Line 169: The authors should include data comparing the number of double positive cells in HLs, HepaRGs and Huh7NTCP o.e. expressing cells under the chosen experimental conditions

We thank the reviewer for this suggestion. To compare the double positive cells among the three cell lines, we co-infected dHepaRGs and Huh7^{NTCP} cells (Revision Figure 5) with HBV and HDV at the same MOI and under the same experimental conditions as HLCs (now Fig 2A, B). We found that dHepaRG also efficiently supported HBV mono-and co-infection, while Huh7^{NTCP} cells were not highly permissive for HBV infection.

Revision Figure 5: HBV/HDV co-infection of dHepaRG cells. Differentiated HepaRG and Huh7-NTCP were infected with HBV (MOI = 450 genome copies/cell) and HDV (MOI = 10) respectively. Cells were stained against HBV core (HBc, magenta), HDAg (green), and nuclei (DAPI, blue) ten days p.i.. HBc- and HDAg-positive cells were counted using Cellprofiler imaging software to quantify HBV and HDV single and co-infection events. Images are representative of three independent differentiations.

Line 217: the complete inhibition of cell to cell spread by myrcludex suggests that there is no spread by cell-cell contact. This should be discussed.

Yes, there is no evidence of HDV spread by cell-cell contact because, as the reviewer correctly points out, BLV treatment almost completely blocked HDV de novo infection (now Fig 3E). To our knowledge, cell-to-cell spread has not been demonstrated for HDV and we have added this aspect to the discussion (lines 412-413). According to our own studies Zhang et al., 2021/2022, J Hepatol, HDV spreads either extracellularly (which can be blocked by BLV) or by cell division. Since HLC are similar to primary human hepatocytes and do not divide in vitro, we believe that extracellular spread is the predominant mode of spread in HLC. We have added this aspect specifically to the new discussion of the revised manuscript (lines 416-417).

Line 210ff: Is there any evidence for syncytia formation in this system?

This is an interesting question, but we have not observed syncytia formation in HDV-infected HLCs. Since HDV has no glycoproteins, we would not expect syncytia to form upon HDV infection. The cells did further not form syncytia even when they were dense.

Line 291: expression analysis by RT-PCR is not sufficient. It will be important to study by CLSM if the identified factors are really present as proteins and properly localized.

We agree with the reviewer on this comment. Since the quantitative assessment of protein expression via CLSM is difficult, we performed WB analysis of lysates from cells obtained at different stages of HLC differentiation to analyse the protein expression of the factors of interest. As shown in the new Fig EV 3E, the WB analysis generally confirmed the trend of the transcriptome analysis.

Regarding the role of CD63: what is the evidence for a direct role of CD63 for HDV entry-can the authors exclude that CD63 is relevant for targeting other factors to the surface? What is the impact of loss of CD63 on the functionality of the autophagosomal-MVB-EV system in HLCs?

Thank you for sharing this insightful comment. We performed immunofluorescence analysis of cells in which we downregulated CD63 and did not observe any obvious changes in surface NTCP or EGFR expression (new Fig EV5B), thus suggesting a direct role of CD63 in HDV cell entry.

CD63 is a member of the tetraspanin family, which plays a crucial role in the formation, function, and trafficking of vesicles along the autophagosomal-MVB-EV (extracellular vesicle) system. CD63 is critically involved in the biogenesis of exosomes, which are EVs formed within the lumen of multivesicular bodies (MVBs), by helping to sort cargo into intraluminal vesicles (ILVs) that eventually become exosomes. CD63 also facilitates the sorting and loading of specific proteins, lipids, and RNAs into exosomes by acting as a scaffold, interacting with various proteins and helping to organise the cargo within the vesicles. Finally, CD63 is involved in autophagy, particularly in the maturation of autophagosomes and their fusion with lysosomes. In addition, MVBs are considered to be a specific type of late endosome (Fader and Colombo, 2009, Cell Death Differ).

Many viruses use late endosomes or lysosomes for cell entry by fusion with their respective membranes. For example, although neither the precise site of HBV viral fusion nor the cues that induce fusion are fully understood, studies suggest that HBV can be co-transported with EGFR and NTCP to late endosomes for trafficking (Herrscher et al; 2020 Cells). We speculate that similar to what has been described for Lujo virus (Tominaga et al, 2014 Molecular cancer), CD63 may be involved in either HDV trafficking and/or virus fusion in the late endosome or lysosome and that downregulation of CD63 negatively affects this process. We have included this aspect in the discussion section of the revised manuscript (lines 474-476).

Minor points:

Line 42: secrete should be replaced by release

We thank the reviewer for pointing out the inaccuracy in our terminology. We have replaced "secrete" with "release" (line 42).

Line 241: proteins are not expressed, genes are expressed

Thank you, we agree and changed the wording accordingly (lines 295-296)

Dear Dr. Dao Thi,

Thank you for the submission of your revised manuscript to our editorial offices. I have now received the reports from the three referees that I asked to re-evaluate the study, you will find below. As you will see, the referees now fully support the publication of the study in EMBO reports. Referees #1 and #3 have some suggestions or requests to improve the manuscript, I ask you to address in a final revised manuscript. Please also provide a final p-b-p-response addressing these points.

- Please provide a completed author checklist with your final submission, which you can download from our author guidelines (<https://www.embopress.org/page/journal/14693178/authorguide>). Please insert page numbers in the checklist to indicate where the requested information can be found in the manuscript. The completed author checklist will also be part of the RPF.
- Please remove the synopsis image, the synopsis blurb and the bullet points from the final manuscript text file. Please upload these separately. See also the instruction for these at the end.
- Please order the manuscript sections like this, using these names:
Title page - Abstract - Keywords - Introduction - Results - Discussion - Methods - Data availability section - Acknowledgements - Disclosure and Competing Interests Statement - References - Figure legends - Expanded View Figure legends
- We now use CRediT to specify the contributions of each author in the journal submission system. CRediT replaces the author contribution section. Please use the free text box to provide more detailed descriptions and do NOT provide your final manuscript text file with an author contributions section. See also our guide to authors:
<https://www.embopress.org/page/journal/14693178/authorguide#authorshipguidelines>
- Please make sure that the number "n" for how many independent experiments were performed, their nature (biological versus technical replicates), the bars and error bars (e.g. SEM, SD) and the test used to calculate p-values is indicated in the respective figure legends. Please also check that all the p-values are explained in the legend, and that these fit to those shown in the figure. Please provide statistical testing where applicable. Please avoid the phrase 'independent experiment', but clearly state if these were biological or technical replicates. Please also indicate (e.g. with n.s.) if testing was performed, but the differences are not significant. In case n=2, please show the data as separate datapoints without error bars and statistics. See also:
<http://www.embopress.org/page/journal/14693178/authorguide#statisticalanalysis>

If n<5, please show single datapoints for diagrams. It seems for many diagrams no error bars are displayed (especially in the main figures). Please add these to all diagrams with n>2 and define these in the legend. Moreover:

- Please note that the exact p values are not provided in the legends of figures 1e-g; 3d-e, h-i; 5b-d, h; EV 2b-c.
- Please indicate the statistical test used for data analysis in the legends of figures 1e; EV 3a.
- Please note that in figures 1e-g; 3h-i; 5b-d; there is a mismatch between the annotated p values in the figure legend and the annotated p values in the figure file that should be corrected.
- Please note that information related to n is missing in the legends of figures 1b-c, e-g, h-j; 2b-f, h; 3b-e, h-i; 5a-d, h; EV 1b-c, f; EV 5a.
- Although 'n' is provided, please describe the nature of entity for 'n' in the legends of figures 4b-d; 5e; EV 1g.
- Please note that the error bars are not defined in the legends of figures 1j; 5a; EV 1b-c, f; EV 2b-c; EV 5a.
- Please note that in figures EV 4c the scale bar unit should be corrected from μM to μm (in the figure legend).
- Please add to each legend (main, EV and Appendix figures, where applicable) a 'Data Information' section explaining the statistics used or providing information regarding replicates and scales. See:

- Please add scale bars of similar style and thickness to all microscopic images (also those in the Appendix), using clearly visible black or white bars (depending on the background). Please place these in the lower right corner of the images themselves. Please do not write on or near the bars in the image but define the size in the respective figure legend. Presently, some scale bars are too thin (see e.g. EV1A and EV1E). Please check.
- Please make sure that all the funding information is also entered into the online submission system and that it is complete and similar to the one in the acknowledgement section of the manuscript text file. Presently, it is 240245660-SFB1129 in the manuscript and 240245660 in the submission system. The BMBF grant VIRASCREEN 01KI2113, and the fellowship from the China Scholarship Council are missing in the submission system. Please check.
- Please remove the table (Table 1) from the manuscript and provide this information in a 'Reagents and Tools' table. I have attached a template for that in word format. Please upload the filled in table to the manuscript tracking system as 'Reagent Table'

file. Please also adjust all callouts for the previous Appendix tables to this table. The example linked below shows how the table will display in the published article and includes examples of the type of information that should be provided for the different categories of reagents and tools. Please list your reagents/tools using the categories provided in the template and do not add additional subheadings to the table. Reagents/tools that do not fit in any of the specific categories can be listed under "Other": https://www.embopress.org/pb%2Dassets/embo-site/msb_177951_sample_FINAL.pdf

- Please remove the referee token from the data availability section and add a link (URL) to directly access the dataset GSE270625. Please make sure that this dataset is public latest on the day of online publication of the manuscript.

- The image in panel 4G (merge) and panel EV4C (left - overlay) look very similar. Please check. If this reuse is intentional, please clearly indicate this in the figure legends.

- Thank you for providing the requested source data. Please upload this as one folder per figure (with all files for one figure in one folder and ZIPed). Moreover, it seems the source data files for 1B and 1C, 2B and 2H and 5E and 5H are identical, which should not be the case. Please check and upload the correct files for these panels for the final submission. Moreover, for 1B (or 1H) the values in column G and H in the first and third row are the same (see attached file). Is that correct? Please check.

In addition, I would need from you uploaded separately:

Best,

Referee #1:

Thank you to the authors for revising their manuscript.

I find that all my concerns were addressed and the manuscript was correctly edited.

I particularly appreciate the effort invested in order to answer questions from Reviewer 2 about the HDV and HBV specific read outs assays. This really makes the manuscript convincing that HBV HDV co infection is possible in HLCs.

Minor comments:

- It seems that the new panel EV3A was not corrected, still showing Go pathways sorted by genes, not significant.
- A few days, a new paper was published identifying NRP1 as a novel host factor for entry of HBV. NRP1 is present in the list of Fig EV3D, and interaction NRP1/CD63 as been described to allow infection by the Arenavirus Lujo Virus. It may be interesting, however not critical, to mention it in the discussion.
- Typo: Page 10, line 375: susecptible

Referee #2:

In the revised version the authors sufficiently addressed all points raised in the previous review process.

Referee #3:

The authors have properly answered my comments. Typos need to be corrected and the quality of the images (Immunofluorescence and western blot) improved.

An additional limitation of the model is the early expression of the large HDAg at similar levels to the S-HDAg resulting in an early inhibition of HDV replication.

In the source data file the authors should indicate the origin of the rest of the samples in the Western blot.

Rev_Com_number: RC-2023-02266
New_manu_number: EMBOR-2024-58873V2
Corr_author: Dao Thi
Title: An Hepatitis B and D Virus Infection Model Using Human Pluripotent Stem Cell-Derived Hepatocytes

Point-by-point reply to the reviewers' comments

Referee #1:

Thank you to the authors for revising their manuscript.

I find that all my concerns were addressed and the manuscript was correctly edited.

I particularly appreciate the effort invested in order to answer questions from Reviewer 2 about the HDV and HBV specific read outs assays. This really makes the manuscript convincing that HBV HDV co infection is possible in HLCs.

Minor comments:

- It seems that the new panel EV3A was not corrected, still showing Go pathways sorted by genes, not significant.
- A few days, a new paper was published identifying NRP1 as a novel host factor for entry of HBV. NRP1 is present in the list of Fig EV3D, and interaction NRP1/CD63 as been described to allow infection by the Arenavirus Lujo Virus. It may be interesting, however not critical, to mention it in the discussion.
- Typo: Page 10, line 375: suseptible

We thank the reviewer again for their very helpful and constructive comments. We apologise for the incorrect EV3A panel and have replaced it with the correct panel showing the pathways in order of significance. Thank you also for drawing our attention to the new study by Yu et al. identifying neuropilin-1 (NRP1) as a novel host factor modulating HBV cell entry. Indeed, as the reviewer pointed out, NRP1 was also upregulated at day 18 of the HLC differentiation protocol, when they became most permissive to HDV infection. Accordingly, we have included NRP1 as a factor in addition to CD63 that could render HLCs highly permissive at this stage (line 324).

Referee #3:

The authors have properly answered my comments. Typos need to be corrected and the quality of the images (Immunofluorescence and western blot) improved.

An additional limitation of the model is the early expression of the large HDAG at similar levels to the S-HDAG resulting in an early inhibition of HDV replication.

In the source data file the authors should indicate the origin of the rest of the samples in the Western blot.

We thank the reviewer for their helpful comments. We have gone through the manuscript thoroughly to remove any typographical errors. We have improved the quality of the images and added the early expression of the large HDAG in the section "Limitations of using HLCs for HBV/HDV infection studies" in the Discussion section of the revised manuscript (lines 476-480).

Dr. Viet Loan Dao Thi
Heidelberg University
Virology
69120
Germany

Dear Dr. Dao Thi,

I am very pleased to accept your manuscript for publication in the next available issue of EMBO reports. Thank you for your contribution to our journal.

Yours sincerely,

Rev_Com_number: RC-2023-02266

New_manu_number: EMBOR-2024-58873V3

Corr_author: Dao Thi

Title: An Hepatitis B and D Virus Infection Model Using Human Pluripotent Stem Cell-Derived Hepatocytes